# NeutrobodyPlex—monitoring SARS-CoV-2 neutralizing immune responses using nanobodies

Teresa R Wagner[1,2,†] (iD), Elena Ostertag[3,†], Philipp D Kaiser[2], Marius Gramlich[2], Natalia Ruetalo[4], Daniel Junker[2], Julia Haering[2], Bjoern Traenkle[2], Matthias Becker[2], Alex Dulovic[2], Helen Schweizer[5] (iD), Stefan Nueske[5], Armin Scholz[5], Anne Zeck[2], Katja Schenke-Layland[2,6,7,8], Annika Nelde[6,9,10], Monika Strengert[11,12], Juliane S Walz[6,9,10,13], Georg Zocher[3], Thilo Stehle[3,14], Michael Schindler[4], Nicole Schneiderhan-Marra[2] & Ulrich Rothbauer[1,2,6,*] (iD)

## Abstract

In light of the COVID-19 pandemic, there is an ongoing need for diagnostic tools to monitor the immune status of large patient cohorts and the effectiveness of vaccination campaigns. Here, we present 11 unique nanobodies (Nbs) specific for the SARS-CoV-2 spike receptor-binding domain (RBD), of which 8 Nbs potently inhibit the interaction of RBD with angiotensin-converting enzyme 2 (ACE2) as the major viral docking site. Following detailed epitope mapping and structural analysis, we select two inhibitory Nbs, one of which binds an epitope inside and one of which binds an epitope outside the RBD:ACE2 interface. Based on these, we generate a biparatopic nanobody (bipNb) with viral neutralization efficacy in the picomolar range. Using bipNb as a surrogate, we establish a competitive multiplex binding assay ("NeutrobodyPlex") for detailed analysis of the presence and performance of neutralizing RBD-binding antibodies in serum of convalescent or vaccinated patients. We demonstrate that NeutrobodyPlex enables high-throughput screening and detailed analysis of neutralizing immune responses in infected or vaccinated individuals, to monitor immune status or to guide vaccine design.

**Keywords** immune response; nanobodies; neutralizing antibodies; SARS-CoV-2; serological assay

**Subject Categories** Immunology; Microbiology, Virology & Host Pathogen Interaction; Structural Biology

## Introduction

As of March 2021, the COVID-19 pandemic has led to the deaths of more than 2.8 million people worldwide and continues to cause severe lockdowns and dramatic economic losses. The emergence and spread of new mutants pose additional risk to current vaccination campaigns (Wang *et al*, 2021). The severe acute respiratory syndrome coronavirus 2 (SARS-CoV-2) as causative agent of the disease expresses a surface spike glycoprotein (spike), which consists of two subunits S1 and S2 (Wrapp *et al*, 2020b; Zhu *et al*, 2020). For viral infection, the receptor-binding domain (RBD), located within S1 interacts with the angiotensin-converting enzyme 2 (ACE2) expressed on human epithelial cells of the respiratory tract (Tai *et al*, 2020; Wrapp *et al*, 2020b; Yan *et al*, 2020). In-depth analysis revealed the presence of spike-specific neutralizing antibodies (NAbs) in convalescent individuals, which were shown to inhibit viral uptake by various mechanisms (Rogers *et al*, 2020; Tortorici *et al*, 2020). In this context, a constantly growing number of NAbs specifically targeting the RBD have been described, underlining the importance of blocking the RBD:ACE2 interaction site for the

1  Pharmaceutical Biotechnology, Eberhard Karls University, Tuebingen, Germany
2  Natural and Medical Sciences Institute, University of Tuebingen, Reutlingen, Germany
3  Interfaculty Institute of Biochemistry, Eberhard Karls University, Tuebingen, Germany
4  Institute for Medical Virology and Epidemiology of Viral Diseases, University Hospital Tuebingen, Tuebingen, Germany
5  Livestock Center of the Faculty of Veterinary Medicine, Ludwig Maximilians University, Oberschleissheim, Germany
6  Cluster of Excellence iFIT (EXC2180) "Image-Guided and Functionally Instructed Tumor Therapies", Eberhard Karls University, Tuebingen, Germany
7  Department of Women's Health, Research Institute for Women's Health, Eberhard Karls University, Tuebingen, Germany
8  Department of Medicine/Cardiology, Cardiovascular Research Laboratories, David Geffen School of Medicine at UCLA, Los Angeles, CA, USA
9  Clinical Collaboration Unit Translational Immunology, German Cancer Consortium (DKTK), Department of Internal Medicine, University Hospital Tuebingen, Tuebingen, Germany
10  Institute for Cell Biology, Department of Immunology, Eberhard Karls University, Tuebingen, Germany
11  Department of Epidemiology, Helmholtz Centre for Infection Research, Braunschweig, Germany
12  TWINCORE GmbH, Centre for Experimental and Clinical Infection Research, A Joint venture of the Hannover Medical School and the Helmholtz Centre for Infection Research, Hannover, Germany
13  Dr. Margarete Fischer-Bosch Institute of Clinical Pharmacology and Robert Bosch Center for Tumor Disease, RBCT, Stuttgart, Germany
14  Vanderbilt University School of Medicine, Nashville, TN, USA
   *Corresponding author. Tel: +49 7121 51530-415; Fax: +49 7121 51530-816; E-mail: ulrich.rothbauer@uni-tuebingen.de
   †These authors contributed equally to this work

development of a protective immune response (Brouwer *et al*, 2020; Cao *et al*, 2020; Ju *et al*, 2020; Robbiani *et al*, 2020; Shi *et al*, 2020; Tai *et al*, 2020; Yu *et al*, 2020). To date, numerous NAbs are in preclinical or clinical development for prophylactic and therapeutic options, providing immediate protection against SARS-CoV-2 infection (reviewed in Jiang *et al*, 2020; Zohar & Alter, 2020).

Promising alternatives to conventional antibodies (IgGs) are single-domain antibodies (nanobodies, Nbs) derived from the heavy-chain antibodies of camelids (Fig 1) (Muyldermans, 2013). Due to their small size and compact fold, Nbs show high chemical stability, solubility, and fast tissue penetration. Nbs can be efficiently selected against different epitopes on the same antigen and converted into multivalent formats (Muyldermans, 2013). The potential of Nbs to address SARS-CoV-2 has been impressively demonstrated by the recent identification of several RBD-specific Nbs from naïve/ synthetic libraries (Chi *et al*, 2020a; Custodio *et al*, 2020; Huo *et al*, 2020; preprint: Schoof *et al*, 2020; preprint: Walter *et al*, 2020; preprint: Ahmad *et al*, 2021) or immunized animals (Chi *et al*, 2020a; preprint: Esparza *et al*, 2020; preprint: Gai *et al*, 2020; Hanke *et al*, 2020; preprint: Nieto *et al*, 2020; Wrapp *et al*, 2020a; Xiang *et al*, 2020; Koenig *et al*, 2021). Some of the identified Nbs show a high viral neutralizing potency proposed by blocking the RBD:ACE2 interface (Custodio *et al*, 2020; Hanke *et al*, 2020; Koenig *et al*, 2021), activation of the SARS-CoV-2 fusion machinery (Koenig *et al*, 2021), or induction of an inactive spike conformation (preprint: Schoof *et al*, 2020).

Since the pandemic outbreak, multiple serological SARS-CoV-2 assays have been established to monitor seroconversion in

individuals and estimate the level of endemic infection in the general population. However, most available serological tests measure the full immune response and can therefore not differentiate between total binding and neutralizing antibodies (Amanat *et al*, 2020; preprint: Lassaunière *et al*, 2020; Robbiani *et al*, 2020; preprint: Roxhed *et al*, 2020; Stadlbauer *et al*, 2020; Tang *et al*, 2020; Becker *et al*, 2021). Detection of the latter is still mostly performed by conventional virus neutralization tests (VNTs), which are both time consuming (2–4 days) and require work with infectious SARS-CoV-2 virions in a specialized biosafety level 3 (BSL3) facility (Muruato *et al*, 2020; Scholer *et al*, 2020).

To overcome these limitations, we aimed to employ Nbs as antibody surrogates and developed a competitive binding approach to screen for neutralizing antibodies on a high-throughput basis in samples from patients or vaccinated individuals. Here, we describe the selection of 11 unique Nbs derived from an alpaca immunized with glycosylated SARS-CoV-2 RBD. Employing a multiplex *in vitro* binding assay, we identified 8 Nbs that effectively block the interaction between RBD, S1, and homotrimeric spike protein (spike) with ACE2 and neutralize SARS-CoV-2 infection in a human cell line. Based on a detailed epitope mapping and structural analysis of RBD: Nb complexes, we selected two of the most potent Nbs simultaneously targeting different epitopes within the RBD and generated a biparatopic Nb (bipNb). The bipNb represents a potent antibody surrogate with $IC_{50}$ values in the low picomolar range and exhibits substantially improved binding affinities. Notably, by addressing at least one conserved epitope outside the RBD:ACE2 interface, the bipNb is still capable to bind recently described RBD mutants derived from strains B.1.1.7 (UK) and B.1.351 (South Africa). To monitor the presence and performance of neutralizing antibodies addressing the RBD:ACE2 interface in convalescent patient samples, we implemented the bipNb in a competitive multiplex binding assay, termed NeutrobodyPlex. Based on the data presented, the NeutrobodyPlex provides a versatile high-throughput approach to screen for a neutralizing immune response in infected or vaccinated individuals, helping to monitor immune status of large populations, to determine the success of vaccination campaigns and to guide vaccine design.

## Results

### Selection of SARS-CoV-2-specific Nbs

To generate Nbs directed against the RBD of SARS-CoV-2, we immunized an alpaca (*Vicugna pacos*) with purified RBD (Amanat *et al*, 2020) and established a Nb phagemid library comprising ~ 4 × 10$^7$ clones representing the full repertoire of variable heavy chains of heavy-chain antibodies ($V_H$Hs or Nbs). After two rounds of phage display on passively adsorbed or biotinylated RBD immobilized on streptavidin plates, we analyzed 492 individual clones in a solid-phase phage ELISA and identified 325 positive binders. Sequence analysis of 72 clones revealed 11 unique Nbs, which cluster into eight families with highly diverse complementarity determining regions (CDR) 3 (Fig 2A, Appendix Table S1). Individual Nbs were produced and purified from *Escherichia coli* (*E. coli*) (Fig 2B), and affinity measurements revealed $K_D$ values ranging from ~ 1.4 to ~ 53 nM indicating the selection of 10 high-affinity monovalent binders. NM1225, that displayed a binding affinity in the micromolar range, was not

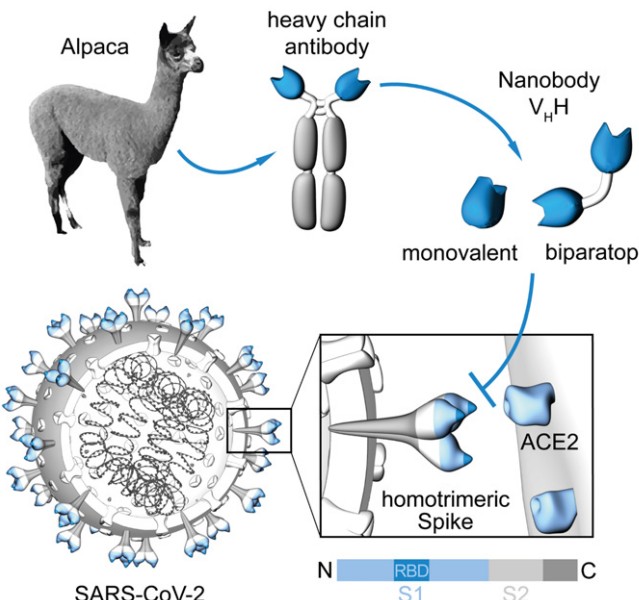

**Figure 1. Generation of nanobodies blocking the SARS-CoV-2 RBD:ACE2 interface.**

Nanobodies (Nbs) are genetically engineered from heavy-chain only antibodies of alpacas. The interaction between the SARS-CoV-2 homotrimeric spike protein and angiotensin-converting enzyme (ACE) 2 can be blocked by receptor-binding domain (RBD)-specific Nbs in the monovalent or biparatopic format.

considered for further analysis (Fig 2C, Appendix Fig S1). Next, we analyzed whether selected Nbs can block the interaction between RBD, S1, or spike of SARS-CoV-2 to human ACE2. To this end, we utilized a multiplex ACE2 competition assay employing the respective SARS-CoV-2 antigens coupled to paramagnetic beads (MagPlex Microspheres) (Becker *et al*, 2021) and incubated them with biotinylated ACE2 and dilutions of purified Nbs, before measuring residual binding of ACE2 via streptavidin-PE conjugate. As controls, we included a non-specific Nb (GFP-Nb, negative control) and two inhibiting mouse antibodies (Gorshkov *et al*, 2020) as positive controls. Data obtained through this multiplex binding assay showed that 8 out of 10 analyzed Nbs inhibit ACE2 binding to all tested SARS-CoV-2 antigens (Fig EV1). Notably, $IC_{50}$ values obtained for the most potent inhibitory Nbs NM1228 (0.5 nM), NM1226 (0.82 nM) and NM1230 (2.12 nM) are highly comparable to $IC_{50}$ values measured for the mouse IgGs (MM43: 0.38 nM; MM57: 3.22 nM).

## Nanobodies show a high potency in neutralizing SARS-CoV-2

Next, a set of Nbs representing the full diversity according to the CDR3 region and highest affinities were examined for their potential to inhibit viral infection. For a viral neutralization test (VNT), human Caco-2 cells were co-incubated with the SARS-CoV-2-mNG

infectious clone and serial dilutions of NM1223, NM1224, NM1226, NM1228, NM1230 or GFP-Nb as negative control. 48 h post-infection neutralization potency was determined via automated fluorescence-microscopy of fixed and nuclear-stained cells. The infection rate, normalized to a non-treated control was plotted and $IC_{50}$ values were determined via sigmoidal inhibition curve fits. NM1226 and NM1228 showed a strong neutralization potency with $IC_{50}$ values of ~ 15 and ~ 7 nM followed by NM1230 (~ 37 nM) and NM1224 (~ 256 nM) (Appendix Fig S2). As expected from our biochemical analysis, NM1223 was not found to neutralize SARS-CoV-2 (Appendix Fig S2). Overall, these findings are highly consistent with the results obtained from the multiplex ACE2 competition assay, thus demonstrating high potencies of ACE2-blocking Nbs to neutralize viral infection.

## Epitope identification

To identify the relative location of Nb epitopes within the RBD, we firstly performed epitope binning experiments using biolayer inter-ferometry (BLI). For this, sensors, pre-coated with biotinylated RBD, were loaded with a first Nb followed by a short dissociation step and subsequent loading of a second Nb (Appendix Fig S3A). As expected, Nbs displaying similar CDR3 sequences (NM1221, NM1222, and NM1230, Nb-Set2) were unable to bind

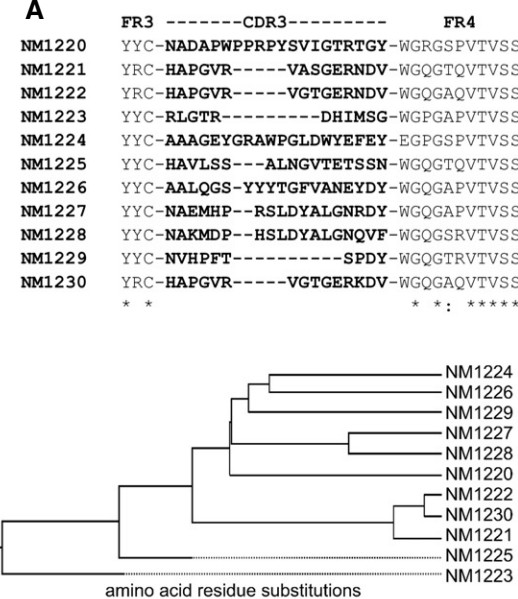

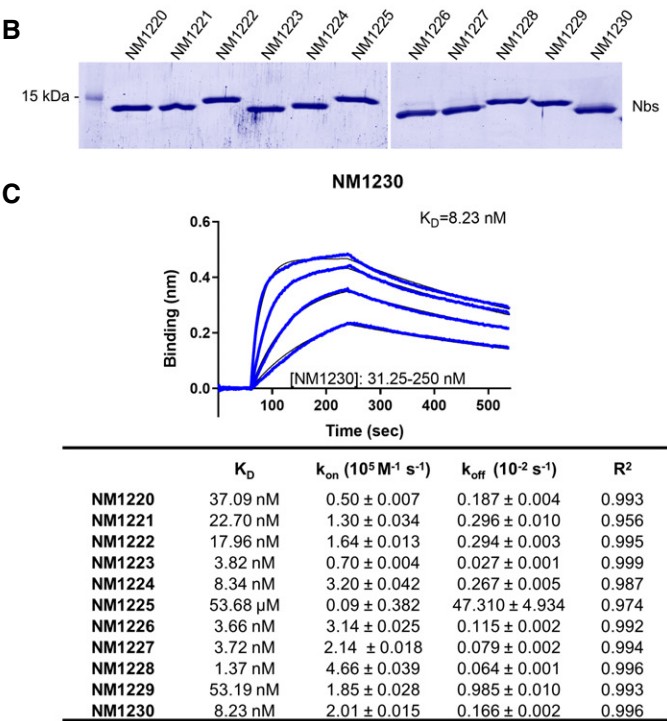

**Figure 2. Biochemical characterization of RBD-specific Nbs.**

A   Amino acid sequences of the complementarity determining region (CDR) 3 from unique Nbs selected after two rounds of biopanning are listed (upper panel). Phylogenetic tree based on a ClustalW alignment of the Nb sequences is shown (lower panel).

B   Recombinant expression and purification of Nbs using immobilized metal affinity chromatography (IMAC) and size exclusion chromatography (SEC). Coomassie staining of 2 μg of purified Nbs is shown.

C   For biolayer interferometry (BLI)-based affinity measurements, biotinylated RBD was immobilized on streptavidin biosensors. Kinetic measurements were performed by using four concentrations of purified Nbs ranging from 15.6 nM to 2 μM. As an example, the sensogram of NM1230 at indicated concentrations is shown (upper panel). The table summarizes affinities ($K_D$), association ($k_{on}$), and dissociation constants ($k_{off}$) determined for individual Nbs (lower panel).

simultaneously as they recognize identical or highly similar epitopes. Interestingly, we noticed that Nbs with highly diverse CDR3s such as NM1228, NM1226, NM1227, and NM1229 could not bind simultaneously, suggesting that these Nbs also recognize similar or overlapping epitopes. Consequently, we clustered these diverse Nbs into Nb-Set1. In total, we identified five distinct Nbs-Sets, comprising at least one candidate targeting a different epitope within the RBD compared with any member of a different Nb-Set (Appendix Fig S3B).

For detailed epitope mapping, we performed Hydrogen-Deuterium Exchange Mass Spectrometry (HDX-MS) using the most potent inhibitory Nbs selected from the different Nb-Sets and determined their binding regions within the RBD in relation to previously described ACE2 interaction sites (Fig EV2A, Appendix Fig S4) (Lan *et al*, 2020; Yan *et al*, 2020). Both members of Nb-Set1, NM1226 and NM1228, interacted with the RBD at the back/ lower right site (Back View, Fig EV2B and C). The highest exchange protection for NM1226 was found in the amino acid (aa) region $N_{RBD}370 - L_{RBD}387$ (Fig EV2B). This region is also covered by NM1228, which displayed additional binding to $Y_{RBD}489 - S_{RBD}514$, being part of the RBD:ACE2 interface (Fig EV2C). NM1230 (Nb-Set2) shows the highest protection in the region of $C_{RBD}432 - L_{RBD}452$ covering two amino acids involved in ACE2 binding ($G_{RBD}446$, $Y_{RBD}449$). A second protected region was found covering $N_{RBD}487 - G_{RBD}496$, which overlaps with the RBD:ACE2 interface (Fig EV2D). In accordance with our binning studies, the main epitope regions differ considerably from both Nb-Sets. As expected, NM1221 and NM1222 (both Nb-Set2) addressed similar RBD regions compared with NM1230 (Fig EV2E and F) while NM1224 (Nb-Set4) showed an interaction distinct from all other Nbs, covering both its main binding region located at the lower right side (Fig EV2G, Front View) and residues in the RBD:ACE2 interface (Fig EV2G, Front View, upper left corner). Notably, the non-inhibitory NM1223 (Nb-Set3) was shown not to contact residues involved in the RBD:ACE2 interface but rather binds to the opposite side (Fig EV2H, Front View).

### Crystal structure of RBD:Nb complexes

To obtain a deeper insight into the binding mode of the most potent inhibitory Nbs, we analyzed the crystal structures of NM1226 and NM1230 in complex with RBD (Fig 3A). All Nb amino acid residues listed below follow the numbering according to Kabat (Kabat *et al*, 1991). The RBD:NM1226 complex structure was solved at a resolution of 2.3 Å (Appendix Table S2). NM1226 was found to interact with various RBD regions (aa 369–384 and aa 504–508) as well as $D_{RBD}405$ and $R_{RBD}408$. Major contacts of NM1226 to RBD are established by residues of the CDR3 ($L_{NM1226}95$, $Q_{NM1226}96$, $G_{NM1226}97$, $S_{NM1226}98$, $Y_{NM1226}99$, $Y_{NM1226}100$, $Y_{NM1226}100a$, $V_{NM1226}100e$, $N_{NM1226}100g$, $E_{NM1226}100h$, $Y_{NM1226}100i$, $D_{NM1226}101$) and CDR1 ($Y_{NM1226}30$). Beside a salt-bridge ($D_{NM1226}101$ to $R_{RBD}408$ and $K_{RBD}378$), various hydrogen bonds were found at contact sites (Fig 3B). In total, the interface of NM1226 buries an area of 698 Å². The RBD:NM1226 complex superimposes well onto spike protein (pdb code: 7KMS) with a Cα-rmsd derivation of 1.7 Å for all three RBD and shows that binding of NM1226 takes place exclusively in the "up" conformation of the spike protein (Fig EV3A and B) (Zhou *et al*, 2020). Although its epitope does not overlap with the binding site of ACE2, our structural data suggest that binding of NM1226

prohibits ACE2 recruitment by steric collision, which would explain its neutralizing effect (Fig EV3A and B).

For the RBD:NM1230 complex, we determined the structure at a resolution of 2.9 Å (Appendix Table S2). In accordance with our HDX-MS analysis, NM1230 binds a distinct three-dimensional epitope located at the opposite site of RBD (Fig 3A). Major contacts were formed by residues of the CDR3 ($P_{NM1230}95$, $R_{NM1230}98$, $T_{NM1230}100a$, $E_{NM1230}100c$, $R_{NM1230}100d$, $K_{NM1230}100e$, $D_{NM1230}101$, $V_{NM1230}102$, $W_{NM1230}103$). In contrast to NM1226, also residues located in the framework regions (FR) including FR1 ($Q_{NM1230}1$), FR2 ($N_{NM1230}35$, $Y_{NM1230}37$, $Q_{NM1230}39$, $G_{NM1230}42$, $K_{NM1230}43$, $A_{NM1230}44$, $L_{NM1230}45$ and $L_{NM1230}47$, and $A_{NM1230}49$), and FR3 ($E_{NM1230}60$) are involved in binding. Consequently, specific binding of NM1230 is achieved by a combination of the CDR3 and residues of the framework regions as previously observed for other Nbs (Kirchhofer *et al*, 2010). On the RBD site, NM1230 interacts with $Y_{RBD}351$ and the large loop region ranging from aa 437–503 (Fig 3C). The RBD:NM1230 interface is mainly formed by polar contacts and one salt-bridge ($R_{NM1230}98$ to $E_{RBD}484$), but also π-π stackings ($W_{NM1230}103$ to $F490_{RBD}$, $R_{NM1230}100d$ to $Y_{RBD}489$) are present (Fig 3C). In total, it buries a surface area of ~ 830 Å². The alignment (Cα-rmsd deviation of 1.2 Å for all three RBD) and comparison to a recently reported structure of the SARS-CoV-2 spike protein bound to neutralizing Nb-Ty1 (pdb code: 6ZXN) (Hanke *et al*, 2020) revealed that NM1230 is able to bind in "up" as well as in "down" conformation (Fig EV3A and B; Appendix Fig S5A).

Next, we compared the RBD:NM1230 structure with the recently reported RBD:ACE2 receptor complex (pdb code:6M17) (Yan *et al*, 2020) to structurally validate the neutralizing potential of NM1230. Closer inspection of the binding site (Appendix Fig S5B and C) showed that NM1230 partially overlaps with the ACE2-binding interface and that the neutralizing effect can be isolated to a limited set of residues within the RBD ($K_{RBD}417$, $Y_{RBD}449$, $F_{RBD}456$, $Y_{RBD}489$, $Q_{RBD}493$, $S_{RBD}494$) (Fig 3C, Appendix Fig S5C). However, NM1230 does not only block binding to ACE2 via its interaction with the RBD on the same protomer (Appendix Fig S5A, clash I), but also impairs ACE2 binding through steric collision on the neighboring RBD (Appendix Fig S5A, clash II). Based upon our structural data, we propose that NM1230-mediated blocking of two out of three RBDs would suffice to abolish ACE2 binding to a trimeric spike molecule.

Considering viral mutagenesis of SARS-CoV-2 within the RBD and associated occurrence of new super spreading strains (preprint: Davies *et al*, 2020; preprint: Tegally *et al*, 2020; Davies *et al*, 2021; Lauring & Hodcroft, 2021; preprint: Walker *et al*, 2021), we analyzed the impact of mutated RBD positions in the recently described SARS-CoV-2 strains B.1.1.7 (UK) and B.1.351 (South Africa) on binding to both neutralizing Nbs NM1226 and NM1230. From our structural data, we assumed that the common amino acid exchange on position 501 (N > Y) has no influence on Nb binding, whereas exchanges on position 417 (K > N) and 484 (E > K), both present in B.1.351, might only affect binding of NM1230 (Appendix Fig S6). In fact, when determining affinities of both Nbs for the RBD mutants derived from B.1.1.7 or B.1.351 by BLI, we obtained similar affinities compared with $RBD_{wt}$ for NM1226 ($K_D$ = ~ 5.4 nM for $RBD_{B.1.1.7}$; $K_D$ = ~ 5.5 nM for $RBD_{B.1.351}$) (Appendix Fig S7A). Interestingly, while affinity of NM1230 to $RBD_{B.1.1.7}$ was not affected ($K_D$ = ~ 10 nM), it decreased for $RBD_{B.1.351}$ ($K_D$ = ~ 26 nM) (Appendix Fig S7B). In accordance with

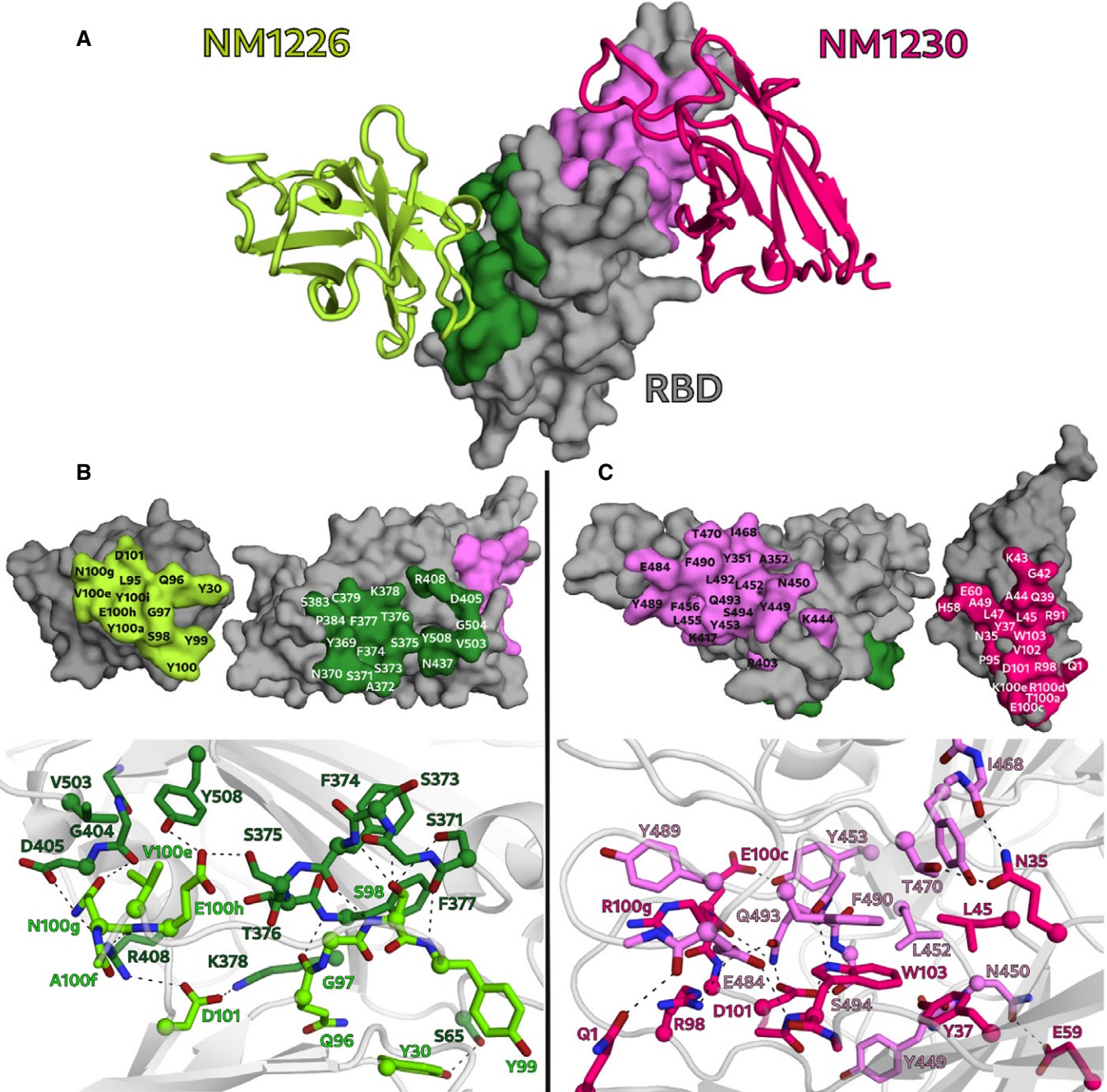

**Figure 3. NM1226 and NM1230 binding epitopes on RBD.**

A Overview of the binding epitopes of NM1226 (light green) on RBD (dark green) and NM1230 (magenta) on RBD (light pink), respectively.

B, C The binding interfaces of NM1226 (B) and NM1230 (C) with the RBD are shown as surface representation (upper panels) and in a close-up view as balls and sticks with direct hydrogen bonds and salt bridges indicated as dotted lines (lower panels). All residues involved in contact formation at a distance cutoff of < 4 Å are listed and labeled. A close-up view of both interfaces reports a detailed interaction map of both contact areas.

our structural epitope analysis, the reduced affinity is most likely due to the loss of the charged interaction between $R_{NM1230}98$ to $E_{RBD}484$. Also, an additional influence by mutation $K_{RBD}417{>}N$ cannot be excluded, since position 417 is part of the NM1230 binding epitope but does not form direct contact to NM1230.

**Generation of a biparatopic Nb with improved efficacies**

Having identified potent Nbs with strong neutralizing characteristics, we proposed that candidates derived from Nb-Set1 (NM1226) and Nb-Set2 (NM1230) might act synergistically. To examine this,

we genetically fused the coding sequences of NM1230 and NM1226 head-to-tail thereby inserting a flexible Gly-Ser linker $((G_4S)_4)$ of 20 amino acids and generated a biparatopic Nb (bipNb) named NM1267. Following production and purification from mammalian cells (Fig 4A), we determined the affinity and analyzed its performance in our multiplex ACE2 competition assay and in the VNT. Compared with the monovalent format, the bipNb showed considerably improved affinities not only for $RBD_{wt}$ ($K_D = \sim 0.5$ nM) but also for RBD mutants ($K_D = \sim 0.6$ nM for $RBD_{B.1.1.7}$; $K_D = \sim 1.15$ nM for $RBD_{B.1.351}$) (Fig 4B and C; Appendix Fig S7C). In line with these findings, it revealed an outstanding inhibition of ACE2 binding to RBD, S1, and spike with an $IC_{50}$ in the low picomolar range (Fig 4B and D). Additionally, we observed an increased potency for viral neutralization indicated by an $IC_{50}$ of $\sim 0.9$ nM (Fig 4B and E), demonstrating that the bipNb NM1267 embodies a substantially refined tool which is beneficial for viral neutralization and competitive binding studies.

## NeutrobodyPlex—using Nbs to determine a SARS-CoV-2 neutralizing immune response

Currently available serological assays provide data on the presence and distribution of antibody subtypes against different SARS-CoV-2 antigens within serum samples of infected and recovered SARS-CoV-2 patients (Amanat *et al*, 2020; preprint: Lassaunière *et al*, 2020; Robbiani *et al*, 2020; preprint: Roxhed *et al*, 2020; Stadlbauer *et al*, 2020; Becker *et al*, 2021; Fink *et al*, 2021). However, they do not differentiate between total and neutralizing RBD-binding antibodies, which sterically inhibit viral entry via ACE2 (Ju *et al*, 2020; Rogers *et al*, 2020; Tang *et al*, 2020). To address this important issue, our bipNb NM1267 might be a suitable surrogate to monitor the emergence and presence of neutralizing antibodies in serum samples of patients. We speculated that NM1267 specifically displaces such antibodies from the RBD:ACE2 interface, which can be monitored as a declining IgG signal (Fig 5A). To test this hypothesis, we first co-incubated antigen-coated beads comprising RBD, S1, or spike with a dilution series of NM1267 and a well-characterized NAb (clone REGN10933), targeting an epitope within the RBD:ACE2 interface (Hansen *et al*, 2020; Jiang *et al*, 2020). As control, we applied the IgG 4A8 (anti-Spike NTD) which was shown to bind an epitope outside the RBD (Chi *et al*, 2020b). Antigen-bound IgGs in the presence of bipNb were detected via an anti-human IgG-PE as mean fluorescent intensities (MFI). Upon addition of increasing concentrations of NM1267, a distinct displacement of REGN10933 was observed (Appendix Fig S8A and B). In contrast, binding of 4A8 was not affected in the presence of NM1267 (Appendix Fig S8C and D), thus proofing the suitability of the bipNb as potent surrogate to displace NAbs targeting the RBD:ACE2 interface. In a next step, we generated a high-throughput competitive binding assay, termed NeutrobodyPlex, by implementing the NM1267 in a recently

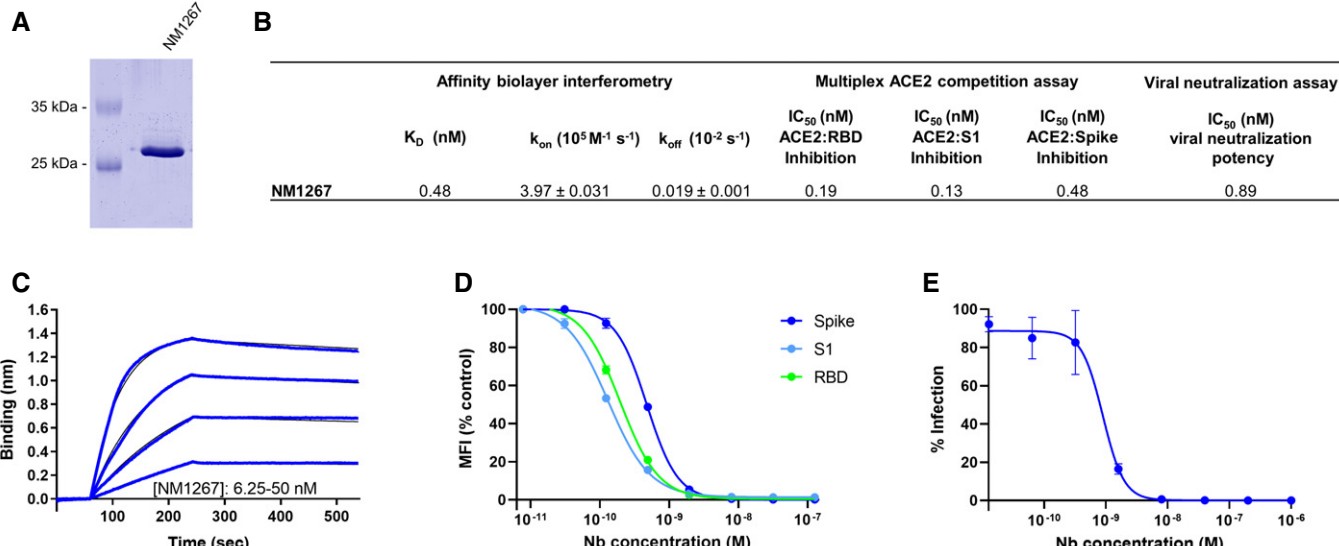

**Figure 4. Biparatopic NM1267 competes with ACE2 and neutralizes SARS-CoV-2 infection.**

A  Coomassie staining of 1 μg of purified biparatopic Nb NM1267.

B  Table summarizing the affinity ($K_D$), association ($k_{on}$), and dissociation constants ($k_{off}$) determined by BLI and $IC_{50}$ values of the multiplex ACE2 competition assay and virus neutralization assay obtained for NM1267.

C  Sensogram of affinity measurements via BLI using four concentrations (6.25–50 nM) of NM1267.

D  Results from multiplex ACE2 competition assay are shown for the three spike-derived antigens: RBD, S1-domain (S1), and homotrimeric spike (Spike). Color-coded beads coated with the respective antigens were co-incubated with biotinylated ACE2 and dilution series of NM1267 (8 pM to 126 nM) followed by measuring residual binding of ACE2. MFI signals were normalized to the maximum detectable signal per antigen given by the ACE2-only control. $IC_{50}$ values were calculated from a four-parametric sigmoidal model. Data are presented as mean ± s.d. of three technical replicates.

E  Neutralization potency of NM1267 was analyzed in Caco-2 cells using the SARS-CoV-2-mNG infectious clones. Infection rate normalized to virus-only infection control is illustrated as percent of infection (% Infection). $IC_{50}$ value was calculated from a four-parametric sigmoidal model, and data are presented as mean ± s.e.m. of three biological replicates ($n = 3$).

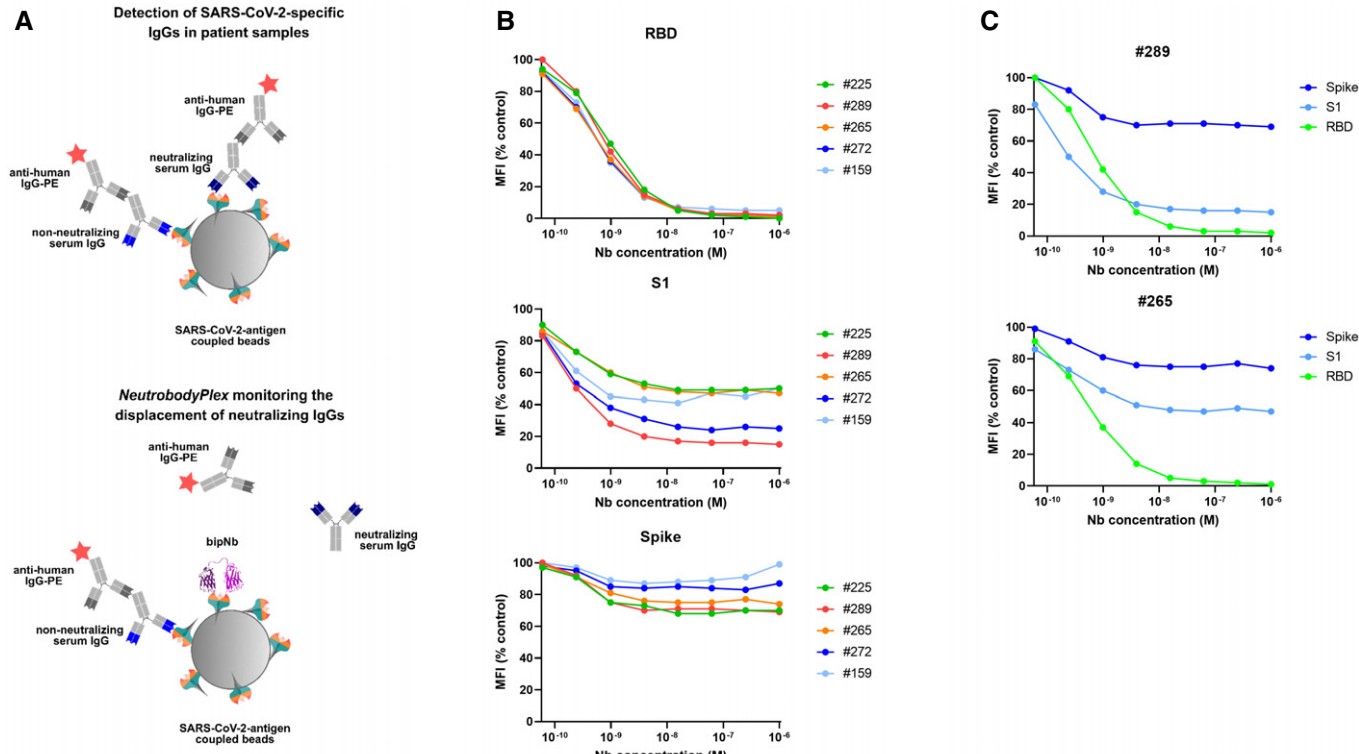

**Figure 5.  NeutrobodyPlex: multiplex competitive binding assay to monitor a neutralizing immune response in patients.**

A   Schematic illustration of the NeutrobodyPlex. The displacement of serum-derived neutralizing IgGs binding to SARS-CoV-2 antigens upon addition of bipNb is measured. In presence of neutralizing IgGs, the fluorescent signal from anti-human-IgG-PE is inversely proportional to the applied bipNb concentration.

B   For the NeutrobodyPlex, antigen-coated beads comprising RBD, S1, or spike were co-incubated with serum samples from five patients and a dilution series of NM1267 (1 μM to 6 pM) (n = 1). Mean fluorescent intensities (MFI) derived from antigen-bound IgGs in the presence of bipNb normalized to the MFI values of IgGs in the serum-only samples, illustrated as MFI (% control), are shown.

C   For two patient-derived serum samples (#289, #265), differences in the NM1267-mediated displacement of IgGs binding the three spike-derived antigens (RBD, S1, spike) are shown.

developed, multiplexed serological assay (Becker *et al*, 2021). Initially, serum samples from five patients were screened. When analyzing IgG binding to RBD, we detected a complete displacement in the presence of ~ 63 nM NM1267. Similarly, a distinct signal reduction for S1 binding IgGs became observable reaching a plateau upon addition of ~ 63 nM NM1267 (Fig 5B, Appendix Table S3). Notably, we observed only minor signal reduction when analyzing spike-binding IgGs (Fig 5B) indicating that the majority of serum IgGs bound this large antigen at epitopes beyond the RBD:ACE2 interaction site (preprint: Heffron *et al*, 2020). From these data, we concluded that all five tested individuals comprise a substantial proportion of neutralizing IgGs that could be detected by competitive bipNb binding using RBD or S1 as antigens. To demonstrate that the NeutrobodyPlex can determine the presence of neutralizing IgGs at detailed resolution, we highlight the results of monitoring the displacement of S1-binding IgGs in two selected patient samples #289 and #265. Upon addition of increasing concentrations of NM1267, we observed a prominent signal reduction of ~ 85 % in patient #289, whereas samples of patient #265 only revealed ~ 53 % displacement of S1-binding IgGs (Fig 5C).

Next, we compared the NeutrobodyPlex on RBD with the cell-based VNT by analyzing a set of 18 serum samples from

convalescent SARS-CoV-2 patients, collected on days 19–57 following a positive PCR test result, and four control samples from healthy donors. To detect differences within the overall immune response and to qualify the neutralizing capacity of the serum samples, we performed the NeutrobodyPlex using two concentrations of NM1267 previously shown to completely (1 μM) or partly (1 nM) displace IgGs binding the RBD:ACE2 interface (Fig 5B). Both assays demonstrated the presence of neutralizing antibodies in all serum samples from convalescent individuals, when using the 1:40 serum dilution in the VNT or when adding 1 μM NM1267 in the NeutrobodyPlex (Fig 6A and B). Notably, neutralizing IgGs were not detected in any of the control serum samples (Appendix Table S4). Further analysis of the displacement of RBD-binding IgGs using the lower NM1267 concentration (1 nM) revealed significant differences in individual neutralization capacities. Whereas some patient samples contained potent neutralizing IgGs that could displace NM1267 upon binding to the RBD:ACE2 interface (Fig 6A, high MFI (% control), light squares), continuous bipNb-mediated displacement of IgGs was detectable in other samples (Fig 6A, low MFI (% control), dark squares), indicating the presence of IgGs with lower neutralizing potency. In parallel, VNT considering the full range of serum dilutions also showed significant differences in individual neutralizing

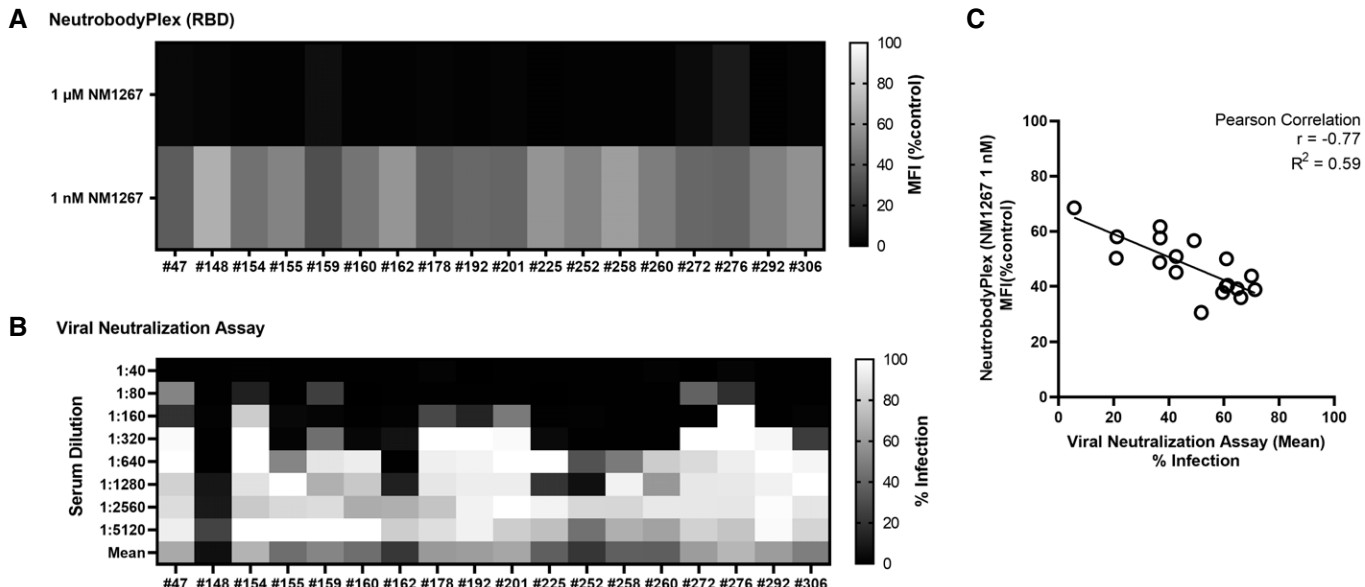

**Figure 6. Specificity of the NeutrobodyPlex in determining individual neutralizing immune responses.**

Serum samples of 18 convalescent SARS-CoV-2-infected individuals were analyzed using the NeutrobodyPlex and the virus neutralization assay.

A   For the NeutrobodyPlex on RBD, two concentrations of NM1267 (1 μM and 1 nM) were applied (*n* = 1). Light colored squares (high MFI (%control)) are indicative for IgGs outcompeting NM1267 from the RBD:ACE2 interface; dark colored squares (low MFI (%control)) show a continuous displacement of IgGs from serum samples in the presence of NM1267.

B   For the virus neutralization assay, serial dilutions of the serum samples (1:40 − 1:5,120) were applied (*n* = 1). Dark colored squares are indicative for a low infection level (low % Infection); light colored squares show a high infection rate (high % Infection).

C   The mean percent of infection (% Infection) derived from all individual serum dilutions obtained by the virus neutralization assay was calculated and plotted against the respective MFI (% control) obtained from the NeutrobodyPlex on RBD in the presence of NM1267 (1 nM). The Pearson correlation determined a negative correlation with $r = -0.77$ and $R^2 = 0.59$ ($P < 0.05$).

potency (Fig 6B). To confirm the validity of the NeutrobodyPlex, we calculated the mean percent of infection (% Infection) of all individual serum dilutions and plotted these values against the respective MFI (% control) obtained from the NeutrobodyPlex (Fig 6C). The observed negative correlation (high MFI (% control) vs. low mean % Infection) strongly suggests that the NeutrobodyPlex allows the detection of individual neutralizing immune responses and assessment of their potency.

For final validation, we screened a serum sample set of 112 convalescent SARS-CoV-2-infected and eight uninfected individuals in the NeutrobodyPlex. In addition to RBD, S1, and spike, we included the S2 domain (S2) and the nucleocapsid (N) of SARS-CoV-2 to monitor total immune response. Incubation with the higher amount of NM1267 (1 μM) revealed the presence of neutralizing IgGs in all serum samples from convalescent patients as shown by an effective replacement of IgGs from RBD, S1, and partially from spike, but not from S2 or N (Fig EV4A). In line with findings from the initial patient sample set, differences in the potencies of neutralizing IgGs present in the individual serum samples became detectable upon addition of the lower concentration of NM1267 (1 nM) (Fig EV4B). As observed previously, non-infected individuals did not display any detectable antibody signal for the presented antigens. We further analyzed, if total level of SARS-CoV-2-specific IgGs correlates with the amount of neutralizing IgGs. As a marker for total IgGs we defined MFI values measured for spike-binding IgGs (Fig 7A and B) and N-binding IgGs (Fig 7C and D) and plotted them against normalized MFI values from IgGs binding to RBD in the

presence of both concentrations of NM1267 (1 μM, 1 nM (MFI RBD (% control)) (Fig 7A–D). Overall, we found high variability of total IgGs (MFI: ∼ 3,500 – ∼ 50,000) in tested serum samples. Upon addition of the higher concentration of NM1267 (1 μM), we observed a complete displacement of IgGs binding the RBD:ACE2 interface independent of the amount of total IgG (Fig 7A and C), whereas upon addition of the lower amount of bipNb, substantial differences of the neutralizing capacity of individual patient samples became detectable (Fig 7B and D). Specifically, using the NeutrobodyPlex, we identified several individuals who have low levels of SARS-CoV-2-specific antibodies overall but have strong neutralizing IgGs (Fig 7B and D highlighted with light-blue background). From these data, it can be concluded that the appearance of neutralizing IgGs after infection with SARS-CoV-2 does not correlate with an overall high SARS-CoV-2-specific antibody titer. However, an increased likelihood for a higher neutralizing capacity in patients showing an overall high SARS-CoV-2-specific IgG level also cannot be excluded. In summary, these results demonstrate how the NeutrobodyPlex can not only provide detailed information on the presence of neutralizing antibodies in patient samples, but also allow qualitative and quantitative assessment of the individual immune response.

# Discussion

In this study, we identified a set of 11 high-affinity RBD-specific Nbs derived from an immunized animal. The high success rate in

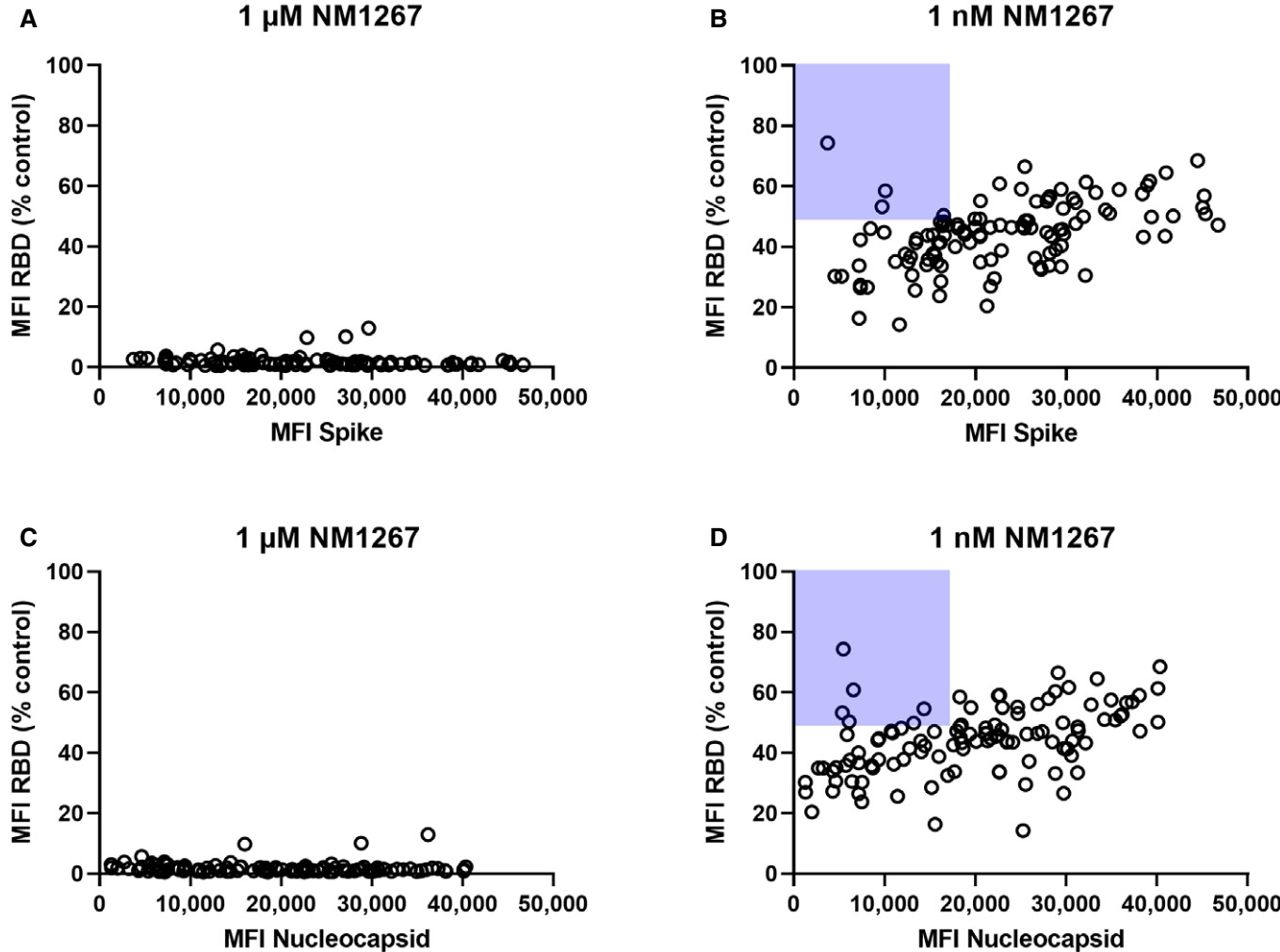

**Figure 7. NeutrobodyPlex enables a differentiated analysis on neutralizing IgGs compared with total SARS-CoV-2 binding IgGs in individuals.**

To investigate neutralizing capacities in relation to total levels of SARS-CoV-2-specific IgGs in individuals, serum samples of 112 convalescent SARS-CoV-2 infected individuals were analyzed using the NeutrobodyPlex (*n* = 1). Individuals displaying strong neutralizing IgGs but showed an overall low level of SARS-CoV-2-specific antibodies are highlighted with light-blue background.

A, B  Total IgGs derived from spike-binding IgGs were plotted against normalized MFI values from IgGs binding to RBD (MFI RBD (%control)) in the presence of both concentrations of NM1267 (1 µM; 1 nM).

C, D  Total IgGs derived from N-binding IgGs were plotted against normalized MFI values from IgGs binding to RBD (MFI RBD (%control)) in the presence of both concentrations of NM1267 (1 µM; 1 nM).

identifying well-functioning Nbs is consistent with recently reported findings describing an ever-growing list of such binders derived from either immunized (Chi *et al*, 2020a; preprint: Esparza *et al*, 2020; preprint: Gai *et al*, 2020; Hanke *et al*, 2020; preprint: Nieto *et al*, 2020; Wrapp *et al*, 2020a; Xiang *et al*, 2020; Koenig *et al*, 2021) or naïve/ synthetic Nb gene libraries (Chi *et al*, 2020a; Custodio *et al*, 2020; Huo *et al*, 2020; preprint: Schoof *et al*, 2020; preprint: Walter *et al*, 2020; preprint: Ahmad *et al*, 2021). By establishing a workflow involving a multiplexed ACE2 competition assay followed by a cell-based VNT, we were able to select high-affinity binding candidates that blocked the interaction between RBD and ACE2 in the context of various SARS-CoV-2 spike-derived antigens and additionally exhibited high viral neutralizing potency. Detailed epitope analysis based on HDX-MS and structural analysis of RBD:

Nb complexes for two of the most potent binders further revealed the molecular basis of the observed neutralizing effect. Both structurally characterized Nbs (NM1226 and NM1230) bind to opposite sides of the RBD. Similar to the recently reported strong neutralizing Nb-Ty1, which displays a partially overlapping binding epitope (Hanke *et al*, 2020), NM1230 can bind to the "down" and "up" form of the spike protomer, thereby inhibiting ACE2 interaction either by steric collisions or due to the overlapping interface (Appendix Fig S5). In contrast, NM1226 binds only the "up" conformation on RBD. As the RBD:ACE2 interface is not addressed by NM1226, its neutralizing effect can most likely be explained due to steric collision. Moreover, NM1226 seems to be more robust against mutations within the RBD:ACE2 interface. Indeed, our comprehensive analyses showed that both Nbs are able to bind the

RBD variant of B.1.1.7 (UK) with similar affinities to $RBD_{wt}$. In comparison, binding of NM1226 to RBD of the more virulent SARS-CoV-2 variant B.1.351 (South Africa) was unaffected, whereas NM1230 showed reduced affinity. To combine the advantages of both single Nbs, add avidity to the construct, and reduce the possibility of viral escape by targeting two independent epitopes, we generated the biparatopic Nb NM1267, which performed excellently in our multiplexed ACE2 competition assay and VNT with $IC_{50}$ values in the picomolar range. In addition, NM1267 revealed high binding affinities to RBDs either derived from B.1.1.7 (UK) or B.1.351 (South Africa) from which we postulated that NM1267 represents a substantially refined tool with highly versatile binding properties.

The ongoing pandemic and the emergence of new mutations leading to more infectious viral strains make the development of improved diagnostic tools and therapies essential. In this context, several studies have shown that humoral immune responses resulting in antibodies directed against the interaction site of RBD and the ACE2 receptor exposed on epithelial cells of the human respiratory tract lead to a strong viral neutralizing effect (Brouwer *et al,* 2020; Cao *et al,* 2020; Chi *et al,* 2020b; Jiang *et al,* 2020; Ju *et al,* 2020; Robbiani *et al,* 2020; Shi *et al,* 2020; Tai *et al,* 2020; Yu *et al,* 2020; Zohar & Alter, 2020). Consequently, there is a keen demand to expand currently applied serological diagnostics toward a more differentiated analysis to specifically monitor and classify immune responses. Therefore, we extended our previously described multiplex serological assay (Becker *et al,* 2021) and implemented NM1267 as a potent surrogate, which specifically displaces IgGs from binding to the RBD:ACE2 interface to monitor the presence and emergence of NAbs in serum samples from convalescent and vaccinated individuals. The NeutrobodyPlex was not only able to successfully detect the overall presence of NAbs targeting the RBD:ACE2 interface, but also to categorize them according to their potency. In addition, it allows comparison between neutralizing IgGs and the total level of SARS-CoV-2 antibodies, which is an important parameter, e.g., for determining the success of a vaccine. However, the NeutrobodyPlex approach is limited in that only NAbs that preferably bind the RBD:ACE2 interaction site can be detected. While such NAbs have been proposed as a milestone for the development of a protective immune response against SARS-CoV-2 (Brouwer *et al,* 2020; Cao *et al,* 2020; Ju *et al,* 2020; Robbiani *et al,* 2020; Shi *et al,* 2020; Tai *et al,* 2020; Yu *et al,* 2020), several NAbs have also been identified that bind outside the RBD:ACE2 interface (Chi *et al,* 2020b; Liu *et al,* 2020), which would be missed by the NeutrobodyPlex.

Compared with recently described assays that directly measure antibody-mediated displacement of ACE2 to determine a neutralizing immune response (preprint: Byrnes *et al,* 2020; Tan *et al,* 2020; Walker *et al,* 2020), a much lower binding affinity of ~ 15–30 nM of RBD to ACE2 must be considered. Consequently, neutralizing immune responses may be detected even in the presence of weak RBD-binding antibodies, which could lead to a false statement of immune protection. Additionally, the usage of small-sized, Nb-derived surrogates further lowers the possibility of a non-targeted and non-reproducible displacement of ACE2, e.g., mediated by steric inhibition and dimerization effects derived from non-specifically binding IgGs. Finally, the multiplex design of the NeutrobodyPlex allows rapid expansion to different antibody isotypes and additional antigens to include, e.g., new relevant RBD mutants that are of importance for advanced vaccination strategies.

To our knowledge, the NeutrobodyPlex demonstrates for the first time a multiplex, antigen-resolved analysis of the presence of neutralizing human IgGs in convalescent individuals suffering from SARS-CoV-2 infection. Compared with other assays, the NeutrobodyPlex can be performed in a fully automated, high-throughput manner and can be readily used for large cohort screening. Since it requires only non-living and non-infectious viral material, costs and safety conditions can be significantly reduced (Muruato *et al,* 2020; Scholer *et al,* 2020; Tan *et al,* 2020). In addition, the NeutrobodyPlex is highly sensitive, as low dilutions of serum (tested dilution: 1:400) are sufficient for analysis, significantly reducing patient material compared with standard assays. However, in its current state, important parameters such as specificity and linearity remain to be defined and more detailed information about the binding properties of neutralizing IgGs are needed. Thus, we have initiated a collaboration with the University Hospital Tuebingen to test large cohorts of convalescent and vaccinated individuals with the NeutrobodyPlex in comparison with RBD ELISA assays and VNTs.

In summary, we envision that the NeutrobodyPlex will provide unique opportunities for detailed classification of individual immune status with respect to the development of protective antibodies. In addition, the NeutrobodyPlex could represent a valuable approach to monitor the efficiency of currently starting vaccination campaigns and their long-term efficacy.

# Materials and Methods

### Expression constructs

For bacterial expression of Nbs, sequences were cloned into the pHEN6 vector (Arbabi Ghahroudi *et al,* 1997), thereby adding a C-terminal $His_6$-Tag for IMAC purification as described previously (Rothbauer *et al,* 2008; Kirchhofer *et al,* 2010). To generate the biparatopic Nb NM1267 from NM1230 and NM1226, cDNAs were amplified by PCR using forward primer 1230Nfor 5´- GGA CGT CTC AAC TCT CAA GTG CAG CTG GTG GAG TC - 3´ and reverse primer 1230Nrev 5´- CAC CAC CGC CAG ATC CAC CGC CAC CTG ATC CTC CGC CTC CTG AGG ACA CGG TGA CCT GGG CCC - 3´ for NM1230 Nb, and forward primer 1226Cfor 5´- GGT GGA TCT GGC GGT GGT GGA AGT GGT GGC GGA GGT AGT CAG GTG CAG CTG GTG GAA T - 3´ and reverse primer 1226Crev 5´- GGG GAA TTC AGT GAT GGT GAT GGT GGT GTG AGG ACA CGG TGA CCG GGG CC - 3´ for NM1226 Nb. Next PCR products from Nbs NM1230 and NM1226 were genetically fused via an internal $(G_4S)_4$–linker by fusion PCR using forward primer 1230Nfor and reverse primer 1226Crev. Full-length cDNA was cloned into Esp3I and EcoRI site of pCDNA3.4 expression vector with N-terminal signal peptide (MGWTLVFLFLLSVTAGVHS) for secretory pathway that comprises Esp3I site.

The pCAGGS plasmids encoding the stabilized homotrimeric spike protein and the receptor-binding domain (RBD) of SARS-CoV-2 were kindly provided by F. Krammer (Amanat *et al,* 2020). The cDNA encoding the S1 domain (aa 1 - 681) of the SARS-CoV-2 spike protein was obtained by PCR amplification using the forward primer S1_CoV2-for 5´- CTT CTG GCG TGT GAC CGG - 3´ and reverse

primer S1_CoV2-rev 5´ - GTT GCG GCC GCT TAG TGG TGG TGG TGG TGG TGG GGG CTG TTT GTC TGT GTC TG - 3´ and the full-length SARS-CoV-2 spike cDNA as template and cloned into the XbaI/ NotI-digested backbone of the pCAGGS vector, thereby adding a C-terminal His$_6$-tag. The optimized sequence of the full-length nucleocapsid protein of SARS-CoV-2 with an N-terminal hexahistidine (His$_6$)-tag was obtained by DNA synthesis (Thermo Fisher Scientific) (GenBank accession numbers QHD43423.2) and cloned into the bacterial expression vector pRSET2b (Thermo Fisher Scientific) via NdeI/HindIII restriction sites. RBDs of virus mutants were generated by PCR amplification of fragments from wild-type DNA template followed by fusion PCRs to introduce described mutation N501Y for B.1.1.7 (RBD$_{B.1.1.7}$) and additional mutations K417N and E484K for B.1.351 (RBD$_{B.1.351}$) (https://covariants.org/shared-mutations). Forward primer RBDfor 5´- ATA TCT AGA GCC ACC ATG TTC GTG TTT CTG G - 3´ and reverse primer N501Yrev 5´- CCA CGC CAT ATG TGG GCT GAA AGC CGT AG - 3´ were used for amplification of fragment 1, forward primer N501Yfor 5´- GGC TTT CAG CCC ACA TAT GGC GTG GGC TAT CAG C - 3´ and reverse primer RBDrev 5´- AAG ATC TGC TAG CTC GAG TCG C - 3´ were used for amplification of fragment 2. Both fragments containing an overlap sequence at the 3` and 5` end were fused by an additional PCR using forward primer RBDfor and RBDrev. Based on cDNA for RBD$_{B.1.1.7}$, additional mutations of B.1.351 were introduced by PCR amplification of three fragments using forward primer RBDfor and reverse primer K417Nrev 5´- GTT GTA GTC GGC GAT GTT GCC TGT CTG TCC AGG G - 3´, forward primer K417Nfor 5´- GAC AGA CAG GCA ACA TCG CCG ACT ACA ACT ACA AGC - 3´ and reverse primer E484Krev 5´- GCA GTT GAA GCC TTT CAC GCC GTT ACA AGG GGT - 3´, forward primer E484Kfor 5´- GTA ACG GCG TGA AAG GCT TCA ACT GCT ACT CCC - 3´ and reverse primer RBDrev. Amplified fragments were assembled by subsequent fusion PCR using forward primer RBDfor and RBDrev. DNA coding for mutant RBDs (amino acids 319–541 of respective spike proteins) was cloned into Esp3I and EcoRI site of pCDNA3.4 expression vector as described for NM1267. All expression constructs were verified by sequence analysis.

**Nb libraries**

Alpaca immunizations with purified RBD and Nb library construction were carried out as described previously (Rothbauer *et al*, 2006). Animal immunization was approved by the government of Upper Bavaria (Permit number: 55.2-1-54-2532.0-80-14). In brief, nine weeks after immunization of an animal (*Vicugna pacos*) with C-terminal histidine-tagged RBD (RBD-His$_6$), ~ 100 ml blood was collected and lymphocytes were isolated by Ficoll gradient centrifugation using the Lymphocyte Separation Medium (PAA Laboratories GmbH). Total RNA was extracted using TRIzol (Life Technologies) and mRNA was reverse transcribed to cDNA using a First-Strand cDNA Synthesis Kit (GE Healthcare). The Nb repertoire was isolated in three subsequent PCRs using the following primer combinations (i) CALL001 (5´-GTC CTG GCT GCT CTT CTA CA A GG-3´) and CALL002 (5´-GGT ACG TGC TGT TGA ACT GTT CC-3´); (ii) forward primer set FR1-1, FR1-2, FR1-3, FR1-4 (5´-CAT GGC NSA NGT GCA GCT GGT GGA NTC NGG NGG-3´, 5´-CAT GGC NSA NGT GCA GCT GCA GGA NTC NGG NGG-3´, 5´-CAT GGC NSA NGT GCA GCT GGT GGA NAG YGG NGG-3´, 5´-CAT GGC NSA NGT GCA

GCT GCA GGA NAG YGG NGG-3´) and reverse primer CALL002 and (3) forward primer FR1-ext1 and FR1-ext2 (5´-GTA GGC CCA GCC GGC CAT GGC NSA NGT GCA GCT GGT GG-3`, 5´-GTA GGC CCA GCC GGC CAT GGC NSA NGT GCA GCT GCA GGA-3´ A-) and reverse primer set FR4-1, FR4-2, FR4-3, FR4-4, FR4-5 and FR4-6 (5`-GAT GCG GCC GCN GAN GAN ACG GTG ACC NGN RYN CC-3´. 5`-GAT GCG GCC GCN GAN GAN ACG GTG ACC NGN GAN CC-3´. 5`-GAT GCG GCC GCN GAN GAN ACG GTG ACC NGR CTN CC-3´. 5`-GAT GCG GCC GCR CTN GAN ACG GTG ACC NGN RYN CC-3´. 5`-GAT GCG GCC GCR CTN GAN ACG GTG ACC NGN GAN CC-3´. 5`-GAT GCG GCC GCR CTN GAN ACG GTG ACC NGR CTN CC-3´) introducing SfiI and NotI restriction sites. The Nb library was subcloned into the SfiI/ NotI sites of the pHEN4 phagemid vector (Arbabi Ghahroudi *et al*, 1997).

**Nb screening**

For the selection of RBD-specific Nbs, two consecutive phage enrichment rounds were performed. TG1 cells containing the "immune"-library in pHEN4 were infected with the M13K07 helper phage, hence the V$_H$H domains were presented superficially on phages. For each round, $1 \times 10^{11}$ phages of the "immune"-library were applied on RBD either directly coated on immunotubes (10 µg/ml) or biotinylated RBD (5 µg/ml) immobilized on 96-well plates pre-coated with Streptavidin. In each selection round, extensive blocking of antigen and phages was performed by using 5% milk or BSA in PBS-T and with increasing panning round, PBS-T washing stringency was intensified. Bound phages were eluted in 100 mM triethylamin, TEA (pH 10.0), followed by immediate neutralization with 1 M Tris/HCl (pH 7.4). For phage preparation for following rounds, exponentially growing TG1 cells were infected and spread on selection plates. Antigen-specific enrichment for each round was monitored by comparing colony number of antigen vs. no antigen selection. Following panning 492 individual clones of the second selection round were screened by standard Phage-ELISA procedures using a horseradish peroxidase-labeled anti-M13 monoclonal antibody (GE Healthcare).

**Protein expression and purification**

RBD-specific Nbs were expressed and purified as previously described (Rothbauer *et al*, 2008; Kirchhofer *et al*, 2010). For the expression of SARS-CoV-2 proteins (RBD, stabilized homotrimeric spike, and S1 domain), Expi293 cells were used (Stadlbauer *et al*, 2020) and the biparatopic Nb NM1267 was expressed using the ExpiCHO system. The full-length nucleocapsid protein (N) was expressed using E.coli BL21 (DE3) cells and purification was performed as described previously (Becker *et al*, 2021). The S2 ecto-domain of the SARS-CoV-2 spike protein (aa 686–1213) was purchased from Sino Biological (cat # 40590, lot # LC14MC3007). For quality control, all purified proteins were analyzed via SDS–PAGE according to standard procedures. For immunoblotting, proteins were transferred on nitrocellulose membranes (Bio-Rad Laboratories) and detection was performed using anti-His antibody (Penta-His Antibody, #34660, Qiagen) followed by donkey-anti-mouse antibody labeled with AlexaFluor647 (Invitrogen) using a Typhoon Trio scanner (GE Healthcare, Freiburg, Germany; excitation 633 nm, emission filter settings 670 nm BP 30).

## Biophysical biolayer interferometry (BLI)

To analyze the binding affinity of purified Nbs toward RBD, biolayer interferometry (BLItz, ForteBio) was performed as per the manufacturer's protocols. Briefly, biotinylated RBD was immobilized on single-use high-precision streptavidin biosensors (SAX). Depending on the affinity of the RBD-Nb interaction, an appropriate concentration range (15.6 nM to 2 μM) of Nbs was used. For each run, four different Nb concentrations were measured as well as a reference run using PBS instead of Nb in the association step. As negative control, GFP-Nb (500 nM) was applied in the binding studies. Global fits were determined using the BLItzPro software and the global dissociation constant ($K_D$) was calculated.

## Bead-based multiplex ACE2 competition assay

Purified RBD, S1 domain, and homotrimeric spike of SARS-CoV-2 were covalently immobilized on spectrally distinct populations of carboxylated paramagnetic beads (MagPlex Microspheres, Luminex Corporation, Austin, TX) using 1-ethyl-3-(3-dimethylaminopropyl)-carbodiimide (EDC)/ sulfo-N-hydroxysuccinimide (sNHS) chemistry as described by Becker et al (2021). Briefly, beads were activated in activation buffer containing 5 mg/ml EDC and 5 mg/ml sNHS. For immobilization, activated beads and respective antigens were incubated in coupling buffer (500 mM MES, pH 5.0, 0.005% (v/v) Triton X-100) for 2 h at 21°C and individual bead populations were combined into a bead mix. For the bead-based ACE2 competition assay, Nbs were incubated with the bead mix (containing beads coupled with SARS-CoV-2 homotrimeric spike, RBD, and S1 proteins) and biotinylated ACE2 (Sino Biological), which competes for the binding of SARS-CoV-2 spike-derived antigens. Single Nbs or bipNbs were pre-diluted to a concentration of 6.3 μmol/L per Nb in assay buffer. Afterward, a fourfold dilution series was made over eight steps in assay buffer containing 160 ng/ml biotinylated ACE2. Subsequently, every dilution was transferred the same volume of bead mix in a 96-well half-area plate. The plate was incubated for 2 h at 21°C, shaking at 750 rpm. Beads were washed using a microplate washer (Biotek 405TS, Biotek Instruments GmbH) to remove unbound ACE2 or Nbs. R-phycoerythrin (PE)-labeled streptavidin was used to detect biotinylated ACE2. Measurements were performed with a FLEXMAP 3D instrument using the xPONENT Software version 4.3 (settings: sample size: 80 μl, 50 events, Gate: 7,500 – 15,000, Reporter Gain: Standard PMT).

## NeutrobodyPlex: Bead-based multiplex neutralizing antibody detection assay

Based on the recently described automatable multiplex immunoassay (Becker et al, 2021), the NeutrobodyPlex was developed and similar assay conditions were applied. For the detection of neutralizing serum antibodies, the bead mix containing beads coupled with purified RBD (receptor-binding domain), S1 (S1 domain), spike (homotrimeric spike), S2 (S2 domain), or N (nucleocapsid) of SARS-CoV-2 was incubated with bipNb NM1267 (concentrations ranging from 1 μM to 6 pM) and purified NAbs (0.08 nM, REGN10933 Cell Sciences; 4A8 ProteoGenix) or serum samples of convalescent SARS-CoV-2 patients and healthy donors at a 1:400 dilution. As positive control and maximal signal detection per sample, serum only was included. Bound serum IgGs were detected via PE-labeled anti-human-IF(ab')₂ Fragment (#109-116-098 Jackson ImmunoResearch) as previously described (Becker et al, 2021).

## Hydrogen-Deuterium exchange

### RBD deuteration kinetics and epitope elucidation

RBD (5 μl, 73 μM) was either incubated with PBS or RBD-specific Nbs (2.5 μl, 2.5 mg/ml in PBS) at 25°C for 10 min. Deuterium exchange of the pre-incubated nanobody-antigen complex was initiated by dilution with 67.5 μl PBS (150 mM NaCl, pH 7.4) prepared with $D_2O$ and incubation for 5 and 50 min, respectively, at 25°C. To ensure a minimum of 90% of complex formation, the molar ratio of antigen to Nbs was calculated as previously described(Kochert et al, 2018), using the affinity constants of 1.37 nM (NM1228), 3.66 nM (NM1226), 3.82 nM (NM1223), 8.23 nM (NM1230), and 8.34 nM (NM1224) (pre-determined by BLI analysis). The final $D_2O$ concentration was 90%. After 5 and 50 min at 25°C, aliquots of 15 μl were taken and quenched by adding 15 μl ice-cold quenching solution (0.2 M TCEP with 1.5% formic acid and 4 M guanidine HCl in 100 mM ammonium formate solution pH 2.2) resulting in a final pH of 2.5. Quenched samples were immediately snap-frozen. The immobilized pepsin was prepared by adding 60 μl of 50% slurry (in ammonium formate solution pH 2.5) to a tube and dried by centrifugation at 1,000 $g$ for 3 min at 0°C and discarding the supernatant. Before injection, aliquots were thawed and added to the dried pepsin beads. Proteolysis was performed for 2 min in a water ice bath followed by filtration using a 22 μm filter and centrifugation at 1,000 $g$ for 30 s at 0°C. Samples were immediately injected into a LC-MS system. Undeuterated control samples were prepared under the same conditions using $H_2O$ instead of $D_2O$. The same protocol was applied for the Nbs without addition of RBD as well to create a list of peptic peptides. The HDX experiments of the RBD-Nb-complex were performed in triplicates. The back-exchange of the method as estimated using a standard peptide mixture of 14 synthetic peptides was 24%.

### Chromatography and mass spectrometry

HDX samples were analyzed on a LC-MS system comprised of RSLC pumps (UltiMate 3000 RSLCnano, Thermo Fisher Scientific, Dreieich, Germany), a chilling device for chromatography (MéCour Temperature Control, Groveland, MA, USA), and a mass spectrometer Q Exactive (Thermo Fisher Scientific, Dreieich, Germany). The chilling device contained the LC column (ACQUITY BEH C18, 1.7 μm, 300 Å, 1 mm × 50 mm (Waters GmbH, Eschborn, Germany)), a cooling loop for HPLC solvents, a sample loop, and the injection valve and kept all components at 0°C. Samples were analyzed using a two-step 20 min linear gradient with a flow rate of 50 μl/min. Solvent A was 0.1% (v/v) formic acid, and solvent B was 80% acetonitrile (v/v) with 0.1% formic acid (v/v). After 3 min desalting at 10% B, a 9 min linear gradient from 10 to 25% B was applied followed by an 8 min linear gradient from 25 to 68.8%. Experiments were performed using a Q Exactive (Thermo Fisher Scientific, Dreieich, Germany) with 70,000 resolutions instrument configurations as follows: sheath gas flow rate of 25; aux gas flow rate of 5; S-lens RF level of 50, spray voltage of 3.5 kV, and a capillary temperature of 300°C.

### HDX data analysis

A peptic peptide list containing peptide sequence, retention time, and charge state was generated in a preliminary LC-MS/MS experiment. The peptides were identified by exact mass and their fragment ion spectrum using protein database searches by Proteome Discoverer v2.1.0.81 (Thermo Fisher Scientific, Dreieich, Germany) and implemented SEQUEST HT search engine. The protein database contained the RBD and the pepsin sequences. Precursor and fragments mass tolerance were set to 6 ppm and 0.05 Da, respectively. No enzyme selectivity was applied; however, identified peptides were manually evaluated to exclude peptides originated through cleavage after arginine, histidine, lysine, proline, and the residue after proline (Hamuro & Coales, 2018). FDR was estimated using q-values calculated by Percolator and only peptides with high-confidence identification (q-value ≤ 0.01) were included to the list. Peptides with overlapping mass, retention time, and charge in Nb and antigen digest, were manually removed. The deuterated samples were recorded in MS mode only, and the generated peptide list was imported into HDExaminer v2.5.0 (Sierra Analytics, Modesto, CA, USA). Deuterium uptake was calculated using the increase in the centroid mass of the deuterated peptides. HDX could be followed for 79% of the RBD amino acid sequence. The calculated percentage deuterium uptake of each peptide between RBD-Nb and RBD-only were compared. Any peptide with uptake reduction of 5% or greater upon Nb binding was considered as protected.

### Crystallization and structural analysis

#### Production of the RBD domain and complex formation with Nbs for crystallization

The RBD domain was produced and purified as described previously (Stadlbauer *et al*, 2020). As a variation to the established protocol, the production of the RBD (residues 319–541) was performed in the Expi293F™ GntI- expression system (Thermo Fisher Scientific, Dreieich, Germany). Expi293F™ GntI- cells were cultivated (37°C, 125 rpm, 8% (v/v) $CO_2$) to a density of $5.5 \times 10^6$ cells/ml. The cells were diluted with Expi293F expression medium to a density of $3.0 \times 10^6$ cells/ml, followed by transfection of RBD plasmid (1 μg/ml cell culture) with Expifectamine (Thermo Fisher Scientific) dissolved in Opti-MEM medium (Thermo Fisher Scientific), according the manufacturer's instructions. 20 h post-transfection, the transfection enhancers were added as documented in the Expi293F™ GntI- cells manufacturer's instructions. The cell suspension was cultivated for 5 days (37°C, 125 rpm, 8% (v/v) $CO_2$) and centrifuged (4°C, 23,900 *g*, 20 min) to clarify the supernatant. The supernatant was supplemented with His-A buffer stock solution (final concentration in the medium: 20 mM $Na_2HPO_4$, 300 mM NaCl, 20 mM imidazole, pH 7.4), before the solution was applied to a HisTrap FF crude column (GE Healthcare). The column was extensively washed with His-buffer-A (20 mM $Na_2HPO_4$, 300 mM NaCl, 20 mM imidazole, pH 7.4) and 50 mM imidazole before the RBD was eluted with 280 mM imidazole from the column. The RBD was dialyzed against PBS and concentrated to 2 mg/ml. The nanobody complex was formed by mixing the RBD with the purified NM1226 or NM1230 in a molar ratio of 1:1.1, followed by incubation for 3 h at 4°C.

For crystallization, the complexes were treated with Endo $H_f$ (New England Biolabs) to truncate the oligosaccharide chain on the RBD. Therefore, Endo $H_f$ (300 U per mg RBD) was added to the RBD:Nb complex and incubated for 2 days at 8°C. Endo $H_f$ was removed by passing the sample through a MBP-Trap column (GE Healthcare). Finally, a size exclusion chromatography using a SD200 16/60 column (GE Healthcare) exchanged the buffer (20 mM HEPES, 150 mM NaCl, pH 7.4) and separated the RBD:Nb complexes from aggregates and nanobody excess.

#### Crystallization

The RBD:NM1230 and RBD:NM1226 complexes were concentrated to 23.2 mg/ml and 29.6 mg/ml, respectively, prior to crystallization. Initial crystallization screening was performed on an ART Robbins Gryphon crystallization robot with placing 400 nL drop RBD:Nb and mixed in a 1:1 ratio with the reservoir solution. For RBD:NM1230, initial crystals appeared overnight in crystallization buffer (200 mM $MgCl_2$, 20% (w/v) PEG 3350, pH 5.9) at 20°C and grew to a final size of $30 \times 30 \times 120$ μm³ within 4 days. After 8 weeks, RBD:NM1226 complex crystallized in a precipitation solution (50 mM $K_2HPO_4$, 20% (w/v) PEG 8000, pH 5.0) at 4°C. The crystals were harvested and frozen in liquid nitrogen until data collection.

#### Structure determination and refinement

Data collection was performed on beamline X06SA at the Swiss Light Source (Villigen, Switzerland). For data reduction, the XDS package was used (Kabsch, 2010) and the resulting dataset was analyzed by XDSSTAT (Diederichs, 2006) to check for radiation damage. For RBD:1226 crystals, a dataset with 2.3 Å resolution was obtained and processed in space group $I4_1$ containing one complex in the asymmetric unit. The tetragonal crystals of RBD:NM1230 diffracted to 2.9 Å and were processed in space group $P4_32_12$ containing two copies of the complex. The structures were solved by molecular replacement using PHASER (McCoy, 2007) and CHAINSAW (Stein, 2008) modified templates of the RBD domain (pdb code: 6Z1Z) and a structure homologue of the nanobody (pdb code: 6XC4). Initial refinement involved simulated annealing as implemented in PHENIX (Adams *et al*, 2010) to reduce model bias. Further refinement was done in a cyclic procedure of reciprocal space refinement as implemented in REFMAC5 (Vagin *et al*, 2004) and real space corrections using COOT (Emsley *et al*, 2010). Several cycles of refinement yielded to a model with good stereochemistry and acceptable R-factors with $R_{work}$/ $R_{free}$ of 26.7% / 30.7% and 19.0% / 22.2% for RBD:NM1230 and RBD:NM1226 complex, respectively (Appendix Table S2). The structure was validated with MOLPROBITY (Williams *et al*, 2018) prior deposition to the protein data bank (pdb code: 7NKT and 7B27 for RBD:NM1226 and RBD:NM1230, respectively). Figures were generated with PYMOL (SCHRODINGER, L. L. C. The PyMOL molecular graphics system. Version, 2010, 1. Jg., Nr. 5, S. 0.) and structure comparison was performed with DALI (Holm, 2019).

### Cell culture

Caco-2 (Human Colorectal adenocarcinoma, ATCC HTB-37) cells were cultured at 37°C with 5% $CO_2$ in DMEM containing 10% FCS, 2 mM l-glutamine, 100 μg/ml penicillin-streptomycin, and 1% NEAA. The cell line was tested negative for mycoplasma using the

PCR mycoplasma kit Venor GeM Classic (Minerva Biolabs) and the Taq DNA Polymerase (Minerva Biolabs).

### Viruses

All experiments associated with the SARS-CoV-2 virus were conducted in Biosafety Level 3 laboratory. The recombinant infectious SARS-CoV-2 clone expressing mNeonGreen (icSARS-CoV-2-mNG) (PMID: 32289263) was obtained from the World Reference Center for Emerging Viruses and Arboviruses (WRCEVA) at the UTMB (University of Texas Medical Branch). To generate icSARS-CoV-2-mNG stocks, 200.000 Caco-2 cells were infected with 50 µl of virus in a 6-well plate; the supernatant was harvested 48 hpi, centrifuged, and stored at −80°C. For MOI determination, a titration using serial dilutions of the mNeonGreen (icSARS-CoV-2-mNG) was conducted. The number of infectious virus particles per ml was calculated as the (MOI × cell number)/ (infection volume), where MOI = −ln(1 − infection rate).

### Virus neutralization assay

For neutralization experiments, $1 \times 10^4$ Caco-2 cells/well were seeded in 96-well plates the day before infection in media containing 5% FCS. Caco-2 cells were co-incubated with the SARS-CoV-2 strain icSARS-CoV-2-mNG at a MOI = 1.1 and Nbs or serum samples in serial dilutions in the indicated concentrations. 48 hpi cells were fixed with 2% PFA and stained with Hoechst33342 (1 µg/ml final concentration) for 10 min at 37°C. The staining solution was removed and exchanged for PBS. For quantification of infection rates, images were taken with the Cytation3 (Biotek) and Hoechst+ and mNG+ cells were automatically counted by the Gen5 Software (Biotek). Infection rate was determined by dividing the number of infected cells through total cell count per condition. Data were normalized to respective virus-only infection control. Inhibitory concentration 50 ($IC_{50}$) was calculated as the half-maximal inhibitory dose using 4-parameter nonlinear regression (GraphPad Prism).

### Patient samples

112 Serum samples of convalescent SARS-CoV-2-infected and eight uninfected individuals were analyzed in the course of this study. All samples used were de-identified and pre-existing. Ethical consent was granted from the Ethics Commission of the University of Tuebingen under the votum 179/2020/BO2. Samples were classified as SARS-CoV-2 infected, based upon a self-reported positive SARS-CoV-2 RT–PCR result.

### Analyses and statistics

Graph preparation and statistical analysis were performed using the GraphPad Prism Software (Version 9.0.0).

## Data availability

Atomic coordinates and structure factors have been deposited in the Protein Data Bank (PDB) under accession code 7NKT (RBD:NM1226 complex, www.rcsb.org/structure/unreleased/7NKT) and 7B27

(RBD:NM1230 complex, www.rcsb.org/structure/unreleased/7B27). Protein sequences of all nanobodies are listed in Appendix Table S1. All data that support the findings of this study are available from the corresponding authors upon reasonable request.

**Expanded View** for this article is available online.

## Acknowledgements

This work was supported by the Initiative and Networking Fund of the Helmholtz Association of German Research Centers (grant number SO-96), the European Union's Horizon 2020 research and innovation program under grant agreement No 101003480—CORESMA. This work has further received funding from State Ministry of Baden-Württemberg for Economic Affairs, Labour and Housing Construction (FKZ 3-4332.62-NMI/68). We thank Florian Krammer for providing expression constructs for SARS-CoV-2 homotrimeric Spike and RBD. Open Access funding enabled and organized by Projekt DEAL.

## Author contributions

Study design: NSM, TRW, MB, UR; Nb selection and biochemical characterization: PDK, BT, TRW; Immunization of the animal: HS, SN, AS; Multiplex binding assay: JH, DJ, MB; HDX-MS experiments: MG, AZ; Organization and providing patient samples: MoS, AN, JSW, KSL; Designing and performing crystallization studies: EO, GZ, TS; Virus neutralization assays: NR, MiS; Data analysis and statistical analysis: TRW, MB, JH, MG, AZ, NR, MiS, UR; Manuscript drafting: TRW, AD, UR; Study supervision: NSM, UR; Manuscript reading: All authors.

## Conflict of interest

T.R.W., P.K., N.S.M., and U.R. are named as inventors on a patent application (EP 20 197 031.6) claiming the use of the described Nanobodies for diagnosis and therapeutics filed by the Natural and Medical Sciences Institute. The other authors declare no competing interest.

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
