## [Review Process File · EMBO Reports]

NeutrobodyPlex - monitoring SARS-CoV-2 neutralizing immune responses using nanobodies

Teresa Wagner, Elena Ostertag, Philipp Kaiser, Marius Gramlich, Natalia Ruetalo, Daniel Junker, Julia Haering, Bjoern Traenkle, Matthias Becker, Alex Dulovic, Helen Schweizer, Stefan Nueske, Armin Scholz, Anne Zeck, Katja Schenke-Layland, Annika Nelde, Monika Strengert, Juliane Walz, Georg Zocher, Thilo Stehle, Michael Schindler, Nicole Schneiderhan-Marra, and Ulrich Rothbauer
DOI: [10.15252/embr.202052325](https://doi.org/10.15252/embr.202052325)

Corresponding author(s): *Ulrich Rothbauer* (ulrich.rothbauer@uni-tuebingen.de)

Review Timeline:

Submission Date:	21st Dec 20
Editorial Decision:	27th Jan 21
Revision Received:	22nd Feb 21
Editorial Decision:	23rd Mar 21
Revision Received:	23rd Mar 21
Accepted:	25th Mar 21

Editor: Achim Breiling

Transaction Report:

Dear Prof. Rothbauer,

Thank you for the submission of your research manuscript to EMBO reports. We have now received the reports from the three referees that were asked to evaluate your study, which can be found at the end of this email.

As you will see, the referees think that these findings are of interest. However, they have several comments, concerns and suggestions, indicating that a major revision of the manuscript is necessary to allow publication of the study in EMBO reports. As the reports are below, and all their points need to be addressed, I will not detail them here.

Given the constructive referee comments, we would like to invite you to revise your manuscript with the understanding that all referee concerns must be addressed in the revised manuscript or in the detailed point-by-point response. Acceptance of your manuscript will depend on a positive outcome of a second round of review. It is EMBO reports policy to allow a single round of revision only and acceptance of the manuscript will therefore depend on the completeness of your responses included in the next, final version of the manuscript.

Revised manuscripts should be submitted within three months of a request for revision. We are aware that many laboratories cannot function at full efficiency during the current COVID-19/SARS-CoV-2 pandemic and we have therefore extended our 'scooping protection policy' to cover the period required for full revision. Please contact me to discuss the revision should you need additional time, and also if you see a paper with related content published elsewhere.

- 1) a .docx formatted version of the final manuscript text (including legends for main figures, EV figures and tables), but without the figures included. Please make sure that changes are highlighted to be clearly visible. Figure legends should be compiled at the end of the manuscript text.
- 2) individual production quality figure files as .eps, .tif, .jpg (one file per figure), of main figures and EV figures. Please upload these as separate, individual files upon re-submission.

The Expanded View format, which will be displayed in the main HTML of the paper in a collapsible format, has replaced the Supplementary information. You can submit up to 5 images as Expanded View. Please follow the nomenclature Figure EV1, Figure EV2 etc. The figure legend for these should be included in the main manuscript document file in a section called Expanded View Figure Legends after the main Figure Legends section. Additional Supplementary material should be supplied as a single pdf file labeled Appendix. The Appendix should have page numbers and needs

to include a table of content on the first page (with page numbers) and legends for all content. Please follow the nomenclature Appendix Figure Sx, Appendix Table Sx etc. throughout the text, and also label the figures and tables according to this nomenclature.

See also our guide for figure preparation:

http://wol-prod-cdn.literatumonline.com/pb-assets/embosite/EMBOPress_Figure_Guidelines_061115-1561436025777.pdf

4) a complete author checklist, which you can download from our author guidelines (<https://www.embopress.org/page/journal/14693178/authorguide>). Please insert page numbers in the checklist to indicate where the requested information can be found in the manuscript. The completed author checklist will also be part of the RPF.

Please also follow our guidelines for the use of living organisms, and the respective reporting guidelines: <http://www.embopress.org/page/journal/14693178/authorguide#livingorganisms>

5) that primary datasets produced in this study (e.g. RNA-seq, ChIP-seq, structural and array data) are deposited in an appropriate public database. If no primary datasets have been deposited, please also state this in the respective section (e.g. 'No primary datasets have been generated and deposited'), see below.

The accession numbers and database should be listed in a formal "Data Availability " section (placed after Materials & Methods) that follows the model below. This is now mandatory (like the COI statement). Please note that the Data Availability Section is restricted to new primary data that are part of this study.

Data availability

- RNA-Seq data: Gene Expression Omnibus GSE46843

(<https://www.ncbi.nlm.nih.gov/geo/query/acc.cgi?acc=GSE46843>)

- [data type]: [name of the resource] [accession number/identifier/doi] ([URL or identifiers.org/DATABASE:ACCESSION])

6) We strongly encourage the publication of original source data with the aim of making primary data more accessible and transparent to the reader. The source data will be published in a separate source data file online along with the accepted manuscript and will be linked to the relevant figure. If you would like to use this opportunity, please submit the source data (for example scans of entire gels or blots, data points of graphs in an excel sheet, additional images, etc.) of your key experiments together with the revised manuscript. If you want to provide source data, please include size markers for scans of entire gels, label the scans with figure and panel number, and send one PDF file per figure.

8) Regarding data quantification and statistics, can you please specify, where applicable, the number "n" for how many independent experiments (biological replicates) were performed, the bars and error bars (e.g. SEM, SD) and the test used to calculate p-values in the respective figure legends. Please provide statistical testing where applicable, and also add a paragraph detailing this to the methods section. See: <http://www.embopress.org/page/journal/14693178/authorguide#statisticalanalysis>

9) Please also note our new reference format:
<http://www.embopress.org/page/journal/14693178/authorguide#referencesformat>

10) Please add a conflict-of-interest statement to the manuscript (below the acknowledgements).

I look forward to seeing a revised version of your manuscript when it is ready. Please let me know if you have questions or comments regarding the revision.

Yours sincerely

Achim Breiling
Editor
EMBO Reports

Referee #1:

In this manuscript by Wagner et al., the authors used a phage display library prepared from an immunized alpaca to identify nanobodies that target the receptor-binding domain (RBD) of SARS-CoV-2. 11 distinct nanobody sequences were identified and characterized by biophysical and SARS-CoV-2 neutralization assays. HDX MS was employed to map RBD epitopes revealing potentially different Nb binding sites. The authors determined a crystal structure of a nanobody NM1230 in complex with the RBD and found that partially overlap with the ACE2 binding sites. In

addition, they developed a bivalent construct by fusing two lead nanobodies that bind non-overlapping epitopes on the receptor-binding motif (RBM). The fusion construct has improved binding affinity and activity.

Based on the hetero-bivalent construct, the authors further expanded the study by developing a competitive assay to characterize serologic activity from COVID convalescent plasma samples. They provided data to imply that this nanobody construct can be used to differentiate COVID positive samples from seronegative controls, potentially providing specificity to neutralizing antibodies that bind specific RBD epitopes that overlap where the two nanobodies interact.

While the lead construct is not among the most potent RBD nanobodies developed to date, this study is comprehensive and provides useful information that adds to the expanding repertoire of COVID nanobodies. Generally, the paper is well written and clear (perhaps less so on the NeutrobodyPlex part). I do have the following concerns that need to be addressed.

Major:

Data transparency: failure to provide X-ray crystal structure information, HDX MS data, and nanobody sequences in the current manuscript have impeded a more comprehensive evaluation of the paper.

I have been struggling to appreciate the novelty of the NeutrobodyPlex assay and its applicability. Would the authors better elaborate on the conceptual advantage(s) of using a single bivalent nanobody v.s. RBD neutralizing IgGs? There are at least 4 or five different high-affinity neutralizing epitopes that have been identified for monoclonal IgGs and can be used to comprehensively evaluate neutralizing epitopes information in patient sera. If the assay is developed for point-of-care and facilitates diagnosis, then it would be useful to compare with other related methods (PCR, CRISPR/Cas9, or RBD ELISA assay). Usually, for detection assay, one might want to show specificity (false positive and false negative), quantification linearity, and sensitivity.

NM1220 seems to bind a distinct epitope from ACE2 binding sites, where does it bind based on HDX?

NM1228 binds RBD strongly (1.37nM) with an epitope that appears to significantly overlap with ACE2. Why did not the authors use this nanobody to design the bivalent form(e.g., presumably replacing more neutralizing IgGs)? Does NM1228 overlap with the critical ACE2 helix for RBD binding?

Specificity: based on the structure, it seems that NM1230 uses many framework residues for RBD binding. Would the specificity of nanobodies be a concern here since frameworks are highly conserved and less specific?

It's not clear what's the fraction of Spike IgGs can bind to the RBD (Fig 8). The authors should show the normalized MFI on the x-axis.

Line 249-250: discrepancy between RBD and spike binding and a claim on Fig 6b that most serum IgGs do not bind the RBD: ACE2 site. Monomeric nanobodies, without IgG-like avidity, may not be able to compete with antibodies for binding of the spike trimer.

Figure S6: data are a bit lousy without proper titrations. For some nanobodies, the IC50s were determined based primarily on one data point and therefore can't be accurately measured.

Minor:

1. It appears that NM1228 represents one of the most potent nanobodies, which was not used for

bioengineering (and X-ray crystallography).

2. Figure 6, can you show the error bar for Figure 6b and 6c?

3. Literature: please expand the published literature on COVID nanobodies that are directly related to the manuscript (there might be more):

Xiang, Y., Nambulli, S., Xiao, Z., Liu, H., Sang, Z., Duprex, W.P., Schneidman-Duhovny, D., Zhang, C., and Shi, Y. (2020). Versatile and multivalent nanobodies efficiently neutralize SARS-CoV-2. *Science* 370, 1479-1484.

Schoof, M., Faust, B., Saunders, R.A., Sangwan, S., Rezelj, V., Hoppe, N., Boone, M., Billesbolle, C.B., Puchades, C., Azumaya, C.M., et al. (2020). An ultrapotent synthetic nanobody neutralizes SARS-CoV-2 by stabilizing inactive Spike. *Science* 370, 1473-1479.

Wrapp, D., De Vlieger, D., Corbett, K.S., Torres, G.M., Wang, N., Van Breedam, W., Roose, K., van Schie, L., Team, V.-C.C.-R., Hoffmann, M., et al. (2020). Structural Basis for Potent Neutralization of Betacoronaviruses by Single-Domain Camelid Antibodies. *Cell* 181, 1436-1441.

4. The GS linker information seems missing.

5. Line 249-250: discrepancy between RBD and spike binding and a claim on Fig 6b that most serum IgGs do not bind the RBD: ACE2 site. Monomeric nanobodies, without IgG-like avidity, may not be able to compete with antibodies for binding of the spike trimer.

Referee #2:

This is an interesting paper, well structured, and novel. The generation of Nanobodies on itself is not novel, nor their characterisation (affinity, epitope binning, neutralisation) and heterodimer production is fairly standard, the authors are translating their affinity reagents towards a multiplex application for screening sera for presence of neutralising antibodies

The report is rather long, but it is difficult to shorten without removing essential information.

In short the paper is suitable for publication in EMBO Reports.

There are only a few minor, minor remarks for the expert judgment of the authors. It is up to them to accept and amend or leave it as it is:

1. They abbreviate their hetero-bivalent Nb as bivNb. While heterodimer Nb is a good description of the construct, normally if two Nbs targeting a different epitope on the same antigen, it is referred to as biparatopic Nb. With 'bivalent Nb', the term they use regularly for their construct, normally refers to the (tandem) homodimer Nb. (e.g. line 100-101)

2. Line 64: "and the current lack of a cure or established vaccine comes with severe lockdowns.." This statement was most likely correct at the time of submission. However, recently, several vaccine have been approved in US, UK, EU and many other countries. So, please rephrase so that the statement becomes timeless.

3. Line 177: the amino acid numbering of Nb is probably the Kabat numbering, please mention this.

4. Line 242-245: The NeutrobodyPlex approach is briefly explained, however, the use of PE labelled streptavidin or anti-human IgG-PE conjugate is missing. Please add this essential component, otherwise the strategy doesn't make sense. (it is found in M&M) . Please mention in M&M that the anti human IgG is a polyclonal goat antibody (this might be important as a minor fraction might also react the VH (like Nbs)).
5. The sentence on line 351-353 is confusing. What do they mean exactly by 'Fc regions of non-specifically binding IgGs.'? is this the Fc of the patient IgG?
6. The sentence on line 353-356: It is evident that by changing to another immobilised RBD mutant, it is possible to investigate whether the patient possess Abs against a particular variant. However, normally we don't know which collection of variants are infecting a victim. So this approach is less relevant, unless it changes dramatically to a new variant throughout the country/nation (like we nearly have in UK). What is more worrying is whether their biparatopic Nb would still bind to there contagious N501Y variant emerging in UK. The authors should not perform extra experiments, however, they could speculate in a singel sentence (around line 356)
7. The lower part of figure 2a could be removed if they CDR3 sequences would be grouped according to their length and AA sequence.
8. In the top panel of figure 2a, the Amino acid upstream of W in FR4 actually belongs to the CDR3 or H3 loop.
9. Line 685: Kon and Koff, the kinetic rate binding, dissociation constants (on or off rates) are written with small 'k'. Capital KD is for equilibrium dissociation constant.

Referee #3:

In this paper, the consortium selected multiple nanobodies from an alpaca immunized with the RBD of the Spike protein. They used in vitro binding assays, epitope mapping, and structural analysis and identified 8 binders, which are able to block the interaction of the Spike protein with ACE2. The design of hetero-bivalent nanobody constructs, targeting different epitopes, significantly improved binding affinities and neutralization efficiencies. Based on these binders, the authors developed a competition assay to screen for neutralizing immune response in infected and vaccinated individuals.

Overall, the experiments were performed properly, the data are well analyzed and interesting, but there are a few major points that need to be addressed or commented on:

- 1.) I'm aware that it is hard to keep up with the speed of similar nanobody/antibody manuscripts being published at the moment, but one should try to keep the reference list up to date. Some studies which are referenced as bioRxiv articles in this article are already published and should be cited correctly (e.g. PMID: 33154106). Furthermore, there are additional studies that are available in bioRxiv (e.g. doi: <https://doi.org/10.1101/2020.11.10.376822>, ...) and not referenced here or even published articles (e.g.: PMID: 33149112), which are not cited.
- 2.) Figure 4 is hard to read and for a non-expert difficult to understand.

3.) I have some conceptual questions/concerns concerning the NeutrobodyPlex assay. To my understanding, this assay is based on the replacement of antibodies (IgGs) by NM1267 from mainly immobilized RBD. Due to the trimeric nature of the RBD in the full-length context (and its different conformations), I believe the Spike protein should be used here because affinities of IgGs might be quite different to RBD or the Spike protein, leading to a misinterpretation of the results.

- Question to Figure 6b: Is it not possible that serum IgGs binds much tighter to Spike (multiple RBDs in close proximity) than NM1267 and therefore the nanobody cannot replace them? This would also challenge the statement that the "majority of serum IgGs bind this large antigen at epitopes beyond the RBD:ACE2 interaction site".

- How can you conclude from the data Fig.6b that the tested individuals "comprise" a substantial fraction of neutralizing IgGs? Does competition with NM1267 automatically mean that they are neutralizing?

I would recommend rerunning the assay by using well-characterized IgGs (known efficient neutralizers with characterized overlapping ACE2 binding epitopes and affinities) on RBD and the Spike protein in the presence of different concentrations of NM1267. Here one would expect that NM1267 is able to replace the IgG from the Spike protein. If this can be shown the assay with all its data would be more convincing.

4.) for the discussion part: How do the selected binders compare to other published nanobody studies (in terms of sequence and binding mode; are they very similar or different?)

5.) What would be the next step of relevance to make use of the study?

Additional comments:

- Abstract line 60: avoid the wording "easily"

- Why was the hetero-bivalent fusion produced in human cells and not E. coli?

- error bars and error estimated are missing for a number of measurements.

Point-by-point response to the Reviewers' Comments

Submission ID: EMBOR-2020-52325V1

MS TITLE: NeutrobodyPlex - Nanobodies to monitor a SARS-CoV-2 neutralizing immune response

We thank all Reviewers for their detailed evaluation of our manuscript and we are pleased by the positive responses of the Reviewers mentioning “... *this study is comprehensive and provides useful information that adds to the expanding repertoire of COVID nanobodies. Generally, the paper is well written and clear.*” (Reviewer 1); “*This is an interesting paper, well structured, and novel. ... The report is rather long, but it is difficult to shorten without removing essential information...In short, the paper is suitable for publication in EMBO Reports*” (Reviewer 2); “*Overall, the experiments were performed properly, the data are well analyzed and interesting*” (Reviewer 3)

We are very grateful for their detailed comments, questions, and suggestions, which help us to present our results in a clearer and more comprehensive fashion. By including new data from additional experiments, we are confident that our revised manuscript now addresses the issues raised by the Reviewers.

Reviewer 1

Major concern:

Data transparency: failure to provide X-ray crystal structure information, HDX MS data, and nanobody sequences in the current manuscript have impeded a more comprehensive evaluation of the paper.

We thank the reviewer for this comment. In the revised manuscript we included the requested information (X-ray crystal structure data of both RBD:Nanobody complexes, HDX MS data and full nanobody sequences), which now can be found either in the main manuscript or as supplementary information in the Appendix. Structure factors and coordinates for both complexes were deposited to the Protein Data Bank and will be available upon release of the manuscript. A validation report of both X-ray structures can be hand in upon request. In addition, we can provide further figures elucidating the quality of electron density, but we cannot hand out coordinates or structure factors prior acceptance of the manuscript.

PDB code for RBD:NM1226 complex: 7NKT

PDB code for RBD:NM1230 complex: 7B27

HDX MS data: Appendix

Amino acid sequences of all nanobodies: **Appendix Table S1**

I have been struggling to appreciate the novelty of the NeurobodyPlex assay and its applicability. Would the authors better elaborate on the conceptual advantage(s) of using a single bivalent nanobody v.s. RBD neutralizing IgGs? There are at least 4 or five different high-affinity neutralizing epitopes that have been identified for monoclonal IgGs and can be used to comprehensively evaluate neutralizing epitopes information in patient sera.

We thank the reviewer for this comment. According to the proposed NeurobodyPlex approach, we demonstrate the use of the high-affinity, biparatopic Nb NM1267 targeting two different epitopes within the RBD as an antibody surrogate, which efficiently can displace RBD specific IgGs present in the serum sample of convalescent individuals. We agree that

previously described neutralizing RBD-binding IgGs (e.g. REGN10933, REGN10987, S309, LY-CoV555, AZD1061 or CTP-59) might be also functional probes to displace unknown RBD-binding IgGs in serum samples. However, as they comprise an Fc part which is needed for detection by a secondary anti-human antibody to monitor changes in IgG binding in the NeutrobodyPlex, such IgGs needs to be reformatted (e.g. into Fab or scFv) and thus are not suitable in their original format for the read out applied in the NeutrobodyPlex.

If the assay is developed for point-of-care and facilitates diagnosis, then it would be useful to compare with other related methods (PCR, CRISPR/Cas9, or RBD ELISA assay). Usually, for detection assay, one might want to show specificity (false positive and false negative), quantification linearity, and sensitivity.

In this manuscript, we present the scientific basis and technical potential of the NeutrobodyPlex to monitor the presence of SARS-CoV-2 neutralizing IgGs in serum samples. Detailed information regarding the specificity and sensitivity of the multiplex binding assay for detecting SARS-CoV-2 binding IgGs in comparison to other serological assays are described in our recent publication on the MULTICOV-Ab (Becker *et al*, 2021). In comparison to current serological assays which are used to determine the overall occurrence of SARS-CoV-2 specific antibodies (Tang *et al*, 2020), the NeutrobodyPlex which classifies IgGs as “neutralizing” based on their potential binding to the RBD:ACE2 interface is not aimed to be applied for point-of-care diagnostics but provide helpful information on the presence of such neutralizing IgGs.

We agree with the reviewer that more data and detailed information are needed before the NeutrobodyPlex can be applied more robustly and reliably to predict a neutralizing immune response. Thus, we have initiated a collaboration with the University Hospital of Tuebingen, where we currently collect data from large convalescent and vaccinated patient cohorts on the emergence of neutralizing antibodies by using the NeutrobodyPlex to validate our

findings. We included this topic more comprehensively in the discussion section of the revised manuscript.

NM1220 seems to bind a distinct epitope from ACE2 binding sites, where does it bind based on HDX?

We decided to perform the HDX-MS epitope analysis for the most potent Nbs with low K_D values. Due to the relatively low affinity ($K_D \sim 37$ nM) of Nb NM1220 we did not include this particular Nb to HDX-MS analysis.

NM1228 binds RBD strongly (1.37nM) with an epitope that appears to significantly overlap with ACE2. Why did not the authors use this nanobody to design the bivalent form (e.g., presumably replacing more neutralizing IgGs)? Does NM1228 overlap with the critical ACE2 helix for RBD binding?

We agree that NM1228 is a very promising candidate due to its high affinity. Indeed we started to generate RBD:NM1228 crystals but unfortunately crystal formation was not successful. Considering our epitope binning and HDX-MS data, we postulated that NM1228 binds a similar/overlapping epitope as NM1226. For NM1226 we successfully performed structural analysis, which we now included in the revised manuscript (**new Figure 3, new Figure EV3, Appendix Table S2**). Nevertheless, we also continued and generated a biparatopic construct combining NM1230 and NM1228. Notably, using the multiplex ACE2 competition assay we obtained highly similar IC_{50} values compared to the biparatopic NM1267, thus we decided not to continue with a second biparatopic Nb construct but focused on NM1267 comprising two Nbs for which we can present detailed structural data.

Specificity: based on the structure, it seems that NM1230 uses many framework residues for RBD binding. Would the specificity of nanobodies be a concern here since frameworks are highly conserved and less specific?

In the past, others and we observed that framework regions of nanobodies participate in antigen binding. For example, one of the most widely used nanobodies, the GFP-Enhancer showed substantial impact of framework 2 on antigen binding ((Kirchhofer *et al*, 2010); PDB: 3K1K) which, however, does not affect specificity. Thus, it is highly conceivable that binding of NM1230 mainly mediated by the elongated interaction between RBD and the CDR3 region provides the observed specificity. However, we address this point shortly in the corresponding result section of the revised manuscript.

It's not clear what's the fraction of Spike IgGs can bind to the RBD (Fig 8). The authors should show the normalized MFI on the x-axis.

We are thankful for this comment and prepared the figure as suggested with a normalized x-axis of the Spike protein.

However, in **new Figure 7** we wanted to highlight the total IgG level (represented by MFI Spike or new MFI Nucleocapsid) compared to neutralizing IgGs detected on RBD (best resolved antigen in the NeutrobodyPlex), therefore we decided not show the normalized MFI on the x-axis. In addition, the proportion of RBD-binding IgGs from spike-binding IgGs cannot be calculated directly because the detected MFI signals do not correspond to antibody amounts. For a broader overview how MFI signals behave upon addition of bipNb, the following box plot illustration was included in this letter.

In our opinion, the new **Figure 7** represents our results in a more meaningful way. Nevertheless, the data could also be included in a revised version of the manuscript as shown here.

Line 249-250: discrepancy between RBD and spike binding and a claim on Fig 6b that most serum IgGs do not bind the RBD: ACE2 site. Monomeric nanobodies, without IgG-like avidity, may not be able to compete with antibodies for binding of the spike trimer.

Here we did not use a monomeric Nb but the biparatopic Nb NM1267. However, to address this question in more detail, we performed additional experiments of the NeutrobodyPlex with well-characterized IgGs instead of patient serum. We could demonstrate that with increasing concentrations of the bipNb, the neutralizing IgG clone REGN10933, which is described to bind to the RBD:ACE2 interface (Hansen *et al*, 2020), can be displaced from RBD, but also from S1 and the homotrimeric spike (new data shown in **Appendix Figure S8A, B** of the

revised manuscript). Furthermore, for RBD we see a complete depletion of RBD-binding serum IgGs in the NeutrobodyPlex, from which it can be assumed that the biparatopic Nb can displace IgGs regardless an increased binding due to IgG derived avidity.

Considering that the RBD represents only a small fraction (~ 150 amino acid residues) of the large spike protein (~ 1500 amino acid residues), it can be assumed that even if the trimeric RBD within the homotrimeric spike protein is completely bound by serum IgGs and the addition of a high excess of up to 1 μ M bipNb displaces all RBD-bound IgGs, this decrease in signal would not be detectable for the large spike protein. In view of this consideration, we assume that the IgG signal detected for spike originates from IgGs binding epitopes outside the RBD:ACE2 site. In addition, it has been previously shown that the RBD:ACE2 binding site is not as immunodominant as expected and that much of the spike-binding antibody targets other portions of the extracellular part of the spike protein (Heffron *et al*, 2020).

Figure S6: data are a bit lousy without proper titrations. For some nanobodies, the IC50s were determined based primarily on one data point and therefore can't be accurately measured.

We agree that the calculation of the IC50 values obtained for the tested individual Nbs in the VNTs (now shown in **Appendix Figure S2**) are not perfect. The IC50-values were obtained from titrating the nanobodies in a range of five serial dilutions and three biological replicates, using triplicate infections in each biological replicate. The dilution range was chosen according to the fact that we expected a breadth of neutralizing response from very high (e.g. NM1228) to rather low (e.g. NM1224). IC50 calculation was done via a four-parametric sigmoidal model without ambiguities. We hence do not feel that this data is “a bit lousy”. Furthermore we like to point out that the VNT was only one of two readouts (multiplex ACE2 competition assay) to determine the neutralization potency of our Nbs, and both assay systems yielded comparable results allowing us to rank the NBs accordingly and to select most potent Nb candidates.

Minor:

It appears that NM1228 represents one of the most potent nanobodies, which was not used for bioengineering (and X-ray crystallography).

We agree that NM1228 is a very promising candidate. Indeed, we started to generate RBD:Nb crystals using this nanobody but unfortunately this was not successful.

Figure 6, can you show the error bar for Figure 6b and 6c?

Data shown in Figure 6 (now **Figure 5** of the revised manuscript) are derived from single experiments measuring signals for ~100 individual color-coded microspheres for each condition and antigen. Optionally, we could include data obtained from a single color coded microsphere as “technical” replicate. Furthermore, by performing a large set of replicates as previously shown for the MULTICOV-Ab, which is based on a highly similar assay format, we obtained standard deviations below 10% (Becker *et al.*, 2021). In line with these findings, we demonstrated in this study that our assays are stable and repeatable (see new **Fig EV1**, standard deviations of maximally 10.1% for the triplicate measurements). Furthermore, for the NeutrobodyPlex we performed serial dilutions of the bipNb, thus major discrepancies would have been visible for the individual dilutions.

Literature: please expand the published literature on COVID nanobodies that are directly related to the manuscript (there might be more):

Xiang, Y., Nambulli, S., Xiao, Z., Liu, H., Sang, Z., Duprex, W.P., Schneidman-Duhovny, D., Zhang, C., and Shi, Y. (2020). Versatile and multivalent nanobodies efficiently neutralize SARS-CoV-2. *Science* 370, 1479-1484.

Schoof, M., Faust, B., Saunders, R.A., Sangwan, S., Rezelj, V., Hoppe, N., Boone, M., Billesballe, C.B., Puchades, C., Azumaya, C.M., et al. (2020). An ultrapotent synthetic nanobody neutralizes SARS-CoV-2 by stabilizing inactive Spike. *Science* 370, 1473-1479.

Wrapp, D., De Vlieger, D., Corbett, K.S., Torres, G.M., Wang, N., Van Breedam, W., Roose, K., van Schie, L., Team, V.-C.C.-R., Hoffmann, M., et al. (2020). Structural Basis for Potent Neutralization of Betacoronaviruses by Single-Domain Camelid Antibodies. *Cell* 181, 1436-1441.

In the revised manuscript, we updated the references substantially. All references listed here are now included.

The GS linker information seems missing

It is now included in the revised manuscript (result section, material and methods)

Reviewer 2

The report is rather long, but it is difficult to shorten without removing essential information

We totally agree and tried to present our data as concisely and short as possible in the revised manuscript.

Minor:

They abbreviate their hetero-bivalent Nb as bivNb. While heterodimer Nb is a good description of the construct, normally if two Nbs targeting a different epitope on the same antigen, it is referred to as biparatopic Nb. With 'bivalent Nb', the term they use regularly for their construct, normally refers to the (tandem) homodimer Nb. (e.g. line 100-101)

We agree and renamed the heterodimer Nb NM1267 now biparatopic (bip) Nb throughout the revised manuscript.

Line 64: "and the current lack of a cure or established vaccine comes with severe lockdowns." This statement was most likely correct at the time of submission. However, recently, several vaccine have been approved in US, UK, EU and many other countries. So, please rephrase so that the statement becomes timeless.

We updated our wording according to the current situation in the revised manuscript.

Line 177 The amino acid numbering of Nb is probably the Kabat numbering, please mention this.

We thank the reviewer for this comment. Indeed we have oversaw to number the Nb sequences accordingly. This is now included in the revised manuscript and properly stated in the result section.

Line 242-245: The NeutrobodyPlex approach is briefly explained, however, the use of PE labelled streptavidin or anti-human IgG-PE conjugate is missing. Please add this essential component, otherwise the strategy doesn't make sense. (it is found in M&M)

We included this information in the result section of the revised manuscript

Please mention in M&M that the anti human IgG is a polyclonal goat antibody (this might be important as a minor fraction might also react the VH (like Nbs). Important info also regarding x reactivity with neutralizing IgGs from human/mouse.

We included this information in the material and methods section of the revised manuscript

The sentence on line 351-353 is confusing. What do they mean exactly by 'Fc regions of non-specifically binding IgGs.'? is this the Fc of the patient IgG?

We changed our statement in the discussion section of the revised manuscript. Now reading:
Additionally, the usage of small-sized, Nb derived surrogates further lowers the possibility of a non-targeted and non-reproducible displacement of ACE2 e.g. mediated by steric inhibition and dimerization effects derived from non-specifically binding IgGs.

The sentence on line 353-356: It is evident that by changing to another immobilised RBD mutant, it is possible to investigate whether the patient possess Abs against a particular variant. However, normally we don't know which collection of variants are infecting a victim. So this approach is less relevant, unless it changes dramatically to a new variant throughout the country/nation (like we nearly have in UK). What is more worrying is whether their biparatopic Nb would still bind to there contagious N501Y variant emerging in UK. The authors should not perform extra experiments, however, they could speculate in a single sentence (around line 356)

With the emergence of new mutations leading to more highly infectious viral strains, we are also greatly interested whether our Nbs and bipNb still can bind these variants. For this reason, we are grateful for the reviewers' suggestion and included novel data concerning mutations present in currently emerging SARS-CoV-2 strains and the binding properties of the Nbs NM1226 and NM1230 (see **Appendix Figure S6** and **Appendix Figure S7** in the revised manuscript).

The lower part of figure 2a could be removed if they CDR3 sequences would be grouped according to their length and AA sequence

To keep the naming of the nanobodies in order but also show their diversity, we would like to keep the tree as shown in the lower part of Figure 2a since the CDR3 are sorted according to their diversity.

In the top panel of figure 2a, the Amino acid upstream of W in FR4 actually belongs to the CDR3 or H3 loop

We thank the reviewer for this comment and adapted the sequences accordingly.

Line 685: k_{on} and k_{off} , the kinetic rate binding, dissociation constants (on or off rates) are written with small 'k'. Capital K_D is for equilibrium dissociation constant.

We are grateful for this comment and corrected the sizing accordingly.

Reviewer 3

Major

I'm aware that it is hard to keep up with the speed of similar nanobody/antibody manuscripts being published at the moment, but one should try to keep the reference list up to date. Some studies which are referenced as bioRxiv articles in this article are already published and should be cited correctly (e.g. PMID: 33154106). Furthermore, there are additional studies that are available in bioRxiv (e.g. doi: <https://doi.org/10.1101/2020.11.10.376822>,) and not referenced here or even published articles (e.g.: PMID: 33149112), which are not cited.

In the revised manuscript, we updated the references substantially. All references listed here are now included.

Figure 4 is hard to read and for a non-expert difficult to understand

For the revision we added new data describing the RBD:NM1226 structure (**new Figure 3**). The new figure including both structures of RBD:NM1226 and RBD:NM1230 was completely restructured and the figure legend rephrased in the revised manuscript accordingly.

I have some conceptual questions/concerns concerning the NeutrobodyPlex assay. To my understanding, this assay is based on the replacement of antibodies (IgGs) by NM1267 from mainly immobilized RBD. Due to the trimeric nature of the RBD in the full-length context (and its different conformations), I believe the Spike protein should be used here because affinities of IgGs might be quite different to RBD or the Spike protein, leading to a misinterpretation of the results

In the NeutrobodyPlex, we conceive in line with the literature that the RBD:ACE2 interface as the most important site addressed by neutralizing antibodies. However, a detailed monitoring of the presence of IgGs binding this particular structure is not possible when using the full length spike protein. As demonstrated, this large antigen is covered by many IgGs, also addressing domains, which are not essential for viral entry such as the S2 domain. Even

IgGs binding conserved epitopes within the RBD but “outside” the ACE2 interface were shown not to exhibit an inhibitory effect (e.g. the CR3022 IgG (Ter Meulen *et al*, 2006);(Yuan *et al*, 2020)).

Question to Figure 6b: Is it not possible that serum IgGs binds much tighter to Spike (multiple RBDs in close proximity) than NM1267 and therefore the nanobody cannot replace them? This would also challenge the statement that the "majority of serum IgGs bind this large antigen at epitopes beyond the RBD:ACE2 interaction site".

To address this question, we performed new experiments and analyzed the displacement of the well-characterized, high affinity and neutralizing IgG clone REGN10933 (Hansen *et al*, 2020) on RBD, S1 and spike by adding our NM1267. By this, we could show that with increasing concentrations of NM1267, REGN10933 is efficiently displaced from all tested antigens (see **Appendix Figure S8**)

Furthermore, for RBD we see a complete depletion of RBD-binding serum IgGs in the NeutrobodyPlex, from which it can be assumed that the biparatopic Nb can displace IgGs regardless an increased binding due to IgG derived avidity.

Considering that the RBD represents only a small fraction (~ 150 amino acid residues) of the large spike protein (~ 1500 amino acid residues), it can be assumed that even if the trimeric RBD within the homotrimeric spike protein is completely bound by serum IgGs and the addition of a high excess of up to 1 μ M bipNb displaces all RBD-bound IgGs, this decrease in signal would not be detectable for the large spike protein. In view of this consideration, we assume that the IgG signal detected for spike originates from IgGs binding epitopes outside the RBD:ACE2 site. In addition, it has been previously shown that the RBD:ACE2 binding site is not as immunodominant as expected and that much of the spike-binding antibody targets other portions of the extracellular part of the spike protein (Heffron *et al*, 2020).

How can you conclude from the data Fig.6b that the tested individuals "comprise" a substantial fraction of neutralizing IgGs? Does competition with NM1267 automatically mean

that they are neutralizing? I would recommend rerunning the assay by using well-characterized IgGs (known efficient neutralizers with characterized overlapping ACE2 binding epitopes and affinities) on RBD and the Spike protein in the presence of different concentrations of NM1267. Here one would expect that NM1267 is able to replace the IgG from the Spike protein. If this can be shown the assay with all its data would be more convincing

We are grateful for the reviewers suggestion to proof our findings that NM1267 can efficiently displace neutralizing IgGs targeting the RBD:ACE2 binding site by using well-characterized IgGs. As shown in **Appendix Figure S8A-D** of the revised manuscript, we applied clone REGN10933 (Hansen *et al.*, 2020) targeting an overlapping epitope on the RBD:ACE2 interface. In parallel we used the anti-Spike-NTD IgG clone 4A8 (Chi *et al.*, 2020) as a control as this IgG binds an epitope outside the RBD:ACE2 interface. By this, we could demonstrate that only REGN10933 was displaced by adding increasing concentration of NM1267 whereas of IgG 4A8 to S1 or spike was not affected.

For the discussion part: How do the selected binders compare to other published nanobody studies (in terms of sequence and binding mode; are they very similar or different?).

In the discussion of the revised manuscript, we included a more detailed comparison to other nanobodies. Furthermore, we discuss more intensively the binding properties of NM1230 in comparison to a recently described neutralizing nanobody (Nb-Ty1, (Hanke *et al.*, 2020), which addresses a similar epitope within the RBD (see **Appendix Figure S5**).

What would be the next step of relevance to make use of the study?

We elaborate on these steps more extensively in the discussion section of the revised manuscript. We are currently employing the NeutrobodyPlex approach to study the presence of neutralizing antibodies in larger cohort of convalescent and vaccinated individuals in collaboration with the University Hospital of Tuebingen.

Abstract line 60: avoid the wording "easily"

done

Why was the hetero-bivalent fusion produced in human cells and not E. coli?

Both systems were tested, however the expression levels of the biparatopic NM1267 was much higher in mammalian cells, where it is produced as a secreting protein. This expression also ensure that all disulfide bridges are correctly formed.

Error bars and error estimated are missing for a number of measurements.

If applicable error bars and error estimations were included. For data of the NeurobodyPlex, most measurements are derived from single experiments measuring signals for ~100 individual color-coded microspheres for each condition and antigen. Optionally, we could include data obtained from a single color coded microsphere as "technical" replicate. However, by performing a large set of replicates as previously shown for the MULTICOV-Ab, which is based on a highly similar assay format, we obtained standard deviations below 10% (Becker *et al.*, 2021). In line with these findings, we demonstrated in this study that our assays are stable and repeatable (see new **Fig EV 1**, standard deviations of maximally 10.1% for the triplicate measurements). Furthermore, for the NeurobodyPlex we performed serial dilutions of the bipNb, thus major discrepancies would have been visible for the individual dilutions.

References:

- Becker M, Strengert M, Junker D, Kaiser PD, Kerrinnes T, Traenkle B, Dinter H, Häring J, Ghozzi S, Zeck A *et al* (2021) Exploring beyond clinical routine SARS-CoV-2 serology using MultiCoV-Ab to evaluate endemic coronavirus cross-reactivity. *Nature Communications* 12: 1152
- Chi X, Yan R, Zhang J, Zhang G, Zhang Y, Hao M, Zhang Z, Fan P, Dong Y, Yang Y *et al* (2020) A neutralizing human antibody binds to the N-terminal domain of the Spike protein of SARS-CoV-2. *Science* 369: 650-655
- Hanke L, Vidakovics Perez L, Sheward DJ, Das H, Schulte T, Moliner-Morro A, Corcoran M, Achour A, Karlsson Hedestam GB, Hallberg BM *et al* (2020) An alpaca nanobody neutralizes SARS-CoV-2 by blocking receptor interaction. *Nat Commun* 11: 4420
- Hansen J, Baum A, Pascal KE, Russo V, Giordano S, Wloga E, Fulton BO, Yan Y, Koon K, Patel K *et al* (2020) Studies in humanized mice and convalescent humans yield a SARS-CoV-2 antibody cocktail. *Science* 369: 1010-1014
- Heffron AS, McIlwain SJ, Baker DA, Amjadi MF, Khullar S, Sethi AK, Shelef MA, O'Connor DH, Ong IM (2020) The landscape of antibody binding to SARS-CoV-2. *bioRxiv*
- Kirchhofer A, Helma J, Schmidthals K, Frauer C, Cui S, Karcher A, Pellis M, Muyldermans S, Casas-Delucchi CS, Cardoso MC (2010) Modulation of protein properties in living cells using nanobodies. *Nature structural & molecular biology* 17: 133
- Tang MS, Case JB, Franks CE, Chen RE, Anderson NW, Henderson JP, Diamond MS, Gronowski AM, Farnsworth CW (2020) Association between SARS-CoV-2 neutralizing antibodies and commercial serological assays. *Clinical Chemistry* 66: 1538-1547
- Ter Meulen J, Van Den Brink EN, Poon LL, Marissen WE, Leung CS, Cox F, Cheung CY, Bakker AQ, Bogaards JA, Van Deventer E (2006) Human monoclonal antibody combination against SARS coronavirus: synergy and coverage of escape mutants. *PLoS Med* 3: e237
- Yuan M, Wu NC, Zhu X, Lee C-CD, So RT, Lv H, Mok CK, Wilson IA (2020) A highly conserved cryptic epitope in the receptor binding domains of SARS-CoV-2 and SARS-CoV. *Science* 368: 630-633

Dear Prof. Rothbauer,

Thank you for the submission of your revised manuscript to our editorial offices. We have now received the reports from the three referees that were asked to re-evaluate your study, you will find below. As you will see, the referees now fully support the publication of your study in EMBO reports.

Before we can proceed with final acceptance, I have these editorial requests I ask you to address in a final revised manuscript:

- Please add up to 5 keywords to the title page (best below the abstract).
- Please provide the abstract written in present tense.
- Maybe, you can update the numbers in the first sentence of the introduction (for March 2021).
- Please order the manuscript sections like this: Title page - Abstract - Introduction - Results - Discussion - Materials and Methods - DAS (data availability section) - Acknowledgements - Author contributions - Conflict of interest - References - Figure legends - Expanded View Figure legends.
- Should the 2 validation reports be seen by the readers? These could be uploaded as dataset then (Dataset EV1 and Dataset EV2) and would need a callout in the manuscript text.
- Figure EV 2 looks a bit fuzzy. Could this be provided with higher resolution?
- Please name the COI "Conflict of interest statement".
- Please make sure that all the funding information is entered also into the online submission system and is complete and similar to the one in the manuscript text file.
- In the author contributions, please change the abbreviation for Michael Schindler to 'Mi.S.' (similar to the callout for Monika Strengert).
- Please call out the separate panels of Fig. EV3 in the manuscript text.
- There is a callout for Supplementary Table 1 on page 28. I guess this should be Appendix Table S1. Please check/change.
- Finally, please find attached a word file of the manuscript text (provided by our publisher) with changes we ask you to include in your final manuscript text, and some queries, we ask you to address. Please provide your final manuscript file with track changes, in order that we can see any modifications done.

In addition, I would need from you:

- a short, two-sentence summary of the manuscript (not more than 40 characters including spaces).
- three - four short bullet points highlighting the key findings of your study
- a schematic summary figure (in jpeg or tiff format with the exact width of 550 pixels and a height of not more than 400 pixels) that can be used as a visual synopsis on our website.

Kind regards,

Achim Breiling
Editor
EMBO Reports

Referee #1:

The main points were properly addressed by the authors. And I support publication of this paper.

Referee #2:

This revised version is ready for acceptance. The authors replied to the comments and critics raised by the reviewers.

Referee #3:

The authors extensively reworked the manuscript on the selection of neutralizing nanobodies against the RBD of the Spike protein. Additional data have been added, certain sections have been rewritten and all the raised concerns have been addressed properly. Well done.

Point-by-point response to the Editorial Comments

Submission ID: EMBOR-2020-52325V2

MS TITLE: NeutrobodyPlex - Nanobodies to monitor a SARS-CoV-2 neutralizing immune response

We thank all Reviewers for their positive response and their assessment that the manuscript is now suitable for publication in EMBO Reports. In the following, we would like to briefly discuss the requested editorial questions / requirements now included in the latest revision

Please add up to 5 keywords to the title page (best below the abstract).

Done

- Please provide the abstract written in present tense.

Done

- Maybe, you can update the numbers in the first sentence of the introduction (for March 2021).

Numbers are updated (March 23rd, 2021)

- Please order the manuscript sections like this: Title page - Abstract - Introduction - Results - Discussion - Materials and Methods - DAS (data availability section) - Acknowledgements - Author contributions - Conflict of interest - References - Figure legends - Expanded View Figure legends.

Order of sections is rearranged as suggested in the revised manuscript

- Should the 2 validation reports be seen by the readers? These could be uploaded as dataset then (Dataset EV1 and Dataset EV2) and would need a callout in the manuscript text.

We think this is not necessary. A full access to our structural data will be available immediately upon publication.

- Figure EV 2 looks a bit fuzzy. Could this be provided with higher resolution?

We added Fig EV2 in higher resolution

- Please name the COI "Conflict of interest statement".

Done

- Please make sure that all the funding information is entered also into the online submission system and is complete and similar to the one in the manuscript text file.

We have also included all indicated funding in the online submission system, which is now coherent with the information in the manuscript

- In the author contributions, please change the abbreviation for Michael Schindler to 'Mi.S.' (similar to the callout for Monika Strengert).

Changed accordingly

- Please call out the separate panels of Fig. EV3 in the manuscript text.

Call outs for Fig EV3A, B are now included

- There is a callout for Supplementary Table 1 on page 28. I guess this should be Appendix Table S1. Please check/change.

Corrected accordingly

- Finally, please find attached a word file of the manuscript text (provided by our publisher) with changes we ask you to include in your final manuscript text, and some queries, we ask you to address. Please provide your final manuscript file with track changes, in order that we can see any modifications done.

We accepted all changes made by the published included them in the revised manuscript. These and our changes in the revised manuscript can be followed by the track changes mode

In addition, I would need from you:

- a short, two-sentence summary of the manuscript (not more than 40 characters including spaces).
- three - four short bullet points highlighting the key findings of your study

Both statements are now included on the first page of the revised manuscript

- a schematic summary figure (in jpeg or tiff format with the exact width of 550 pixels and a height of not more than 400 pixels) that can be used as a visual synopsis on our website.

Suggestion for visual synopsis is included

Prof. Ulrich Rothbauer
Eberhard Karls University Tuebingen
Pharmaceutical Biotechnology
Markwiesenstrasse 55
Reutlingen 72770
Germany

Dear Prof. Rothbauer,

I am very pleased to accept your manuscript for publication in the next available issue of EMBO reports. Thank you for your contribution to our journal.

At the end of this email I include important information about how to proceed. Please ensure that you take the time to read the information and complete and return the necessary forms to allow us to publish your manuscript as quickly as possible.

As part of the EMBO publication's Transparent Editorial Process, EMBO reports publishes online a Review Process File to accompany accepted manuscripts. As you are aware, this File will be published in conjunction with your paper and will include the referee reports, your point-by-point response and all pertinent correspondence relating to the manuscript.

If you do NOT want this File to be published, please inform the editorial office within 2 days, if you have not done so already, otherwise the File will be published by default [contact: emboreports@embo.org]. If you do opt out, the Review Process File link will point to the following statement: "No Review Process File is available with this article, as the authors have chosen not to make the review process public in this case."

Should you be planning a Press Release on your article, please get in contact with emboreports@wiley.com as early as possible, in order to coordinate publication and release dates.

Thank you again for your contribution to EMBO reports and congratulations on a successful publication. Please consider us again in the future for your most exciting work.

Yours sincerely,

Achim Breiling
Editor
EMBO Reports

THINGS TO DO NOW:

You will receive proofs by e-mail approximately 2-3 weeks after all relevant files have been sent to

our Production Office; you should return your corrections within 2 days of receiving the proofs.

Please inform us if there is likely to be any difficulty in reaching you at the above address at that time. Failure to meet our deadlines may result in a delay of publication, or publication without your corrections.

All further communications concerning your paper should quote reference number EMBOR-2020-52325V3 and be addressed to emboreports@wiley.com.

Should you be planning a Press Release on your article, please get in contact with emboreports@wiley.com as early as possible, in order to coordinate publication and release dates.

Corresponding Author Name: Ulrich Rothbauer

Manuscript Number: EMBOR-2020-52325V1